# HIERARCHICAL SPECULATIVE DECODING THROUGH TRAINING-FREE SLIM-VERIFIER

## ABSTRACT

Speculative decoding (SD) addresses the high inference costs of large language models by having lightweight drafters generate candidates for large verifiers to validate in parallel. Current draft-verify methods use binary decisions: accept or fully recompute. We find that current binary verification creates inefficiency: many rejected tokens could be verified correctly with a slim model rather than a full verifier. This motivates our Training-Free Slim-Verifier to handle tokens requiring moderate verification resources, reducing expensive large-model calls. We propose Hierarchical Verification for Speculative Decoding (HVSD), a three-tier training-free framework using a skip-layer slim-verifier. Draft tokens are processed hierarchically: direct acceptance for high-confidence cases, slim-verifier regeneration for medium-confidence cases, and full-model verification for uncertain cases. Across summarization, translation, reasoning, QA, and coding tasks on T5 and Gemma families, HVSD consistently lowers rejection rates (0.1–0.22) and achieves 10–20% speedup over state-of-the-art SDs. Compared to decoding without drafting, HVSD provides 2.5-3× acceleration while improving output quality. Our results establish multi-tier SD as a general paradigm for scalable and efficient LLM inference.

## 1 INTRODUCTION

Due to the high computational cost and latency involved in deploying large models for inference (Patterson, 2004; Hennessy & Patterson, 2012; Shazeer, 2019), researchers have explored methods to improve inference efficiency at the data (Wang & Simoncelli, 2020; Mirzasoleiman et al., 2020; He et al., 2024), model (Shazeer et al., 2017; Han et al., 2016; Hinton et al., 2015), and system levels (Xia et al., 2023; Leviathan et al., 2023; Narayanan et al., 2021). At the system level, speculative decoding (SD) (Burton, 1985; Xia et al., 2023; Leviathan et al., 2023; Zhang et al., 2023) improves the inference efficiency by letting a small model "draft" multiple candidate tokens and only calling the large model to verify them in parallel. In existing draft-verify frameworks, most studies fall into two directions: either enhancing the capability of the drafter (Kim et al., 2023b; Zhou et al., 2023; Liu et al., 2023), or improving the efficiency of verifier (Miao et al., 2023; Cai et al., 2024; Li et al., 2024).

However, the two-tier paradigm suffers from inherent efficiency bottleneck: a token is either directly accepted from the drafter or must be recomputed by the verifier, result-in resulting in excessive large model calls. We find that there exist draft tokens that exceed small drafters' reliable prediction capacity but require only a subset of large verifiers' computational resources for accurate verification.Yet in such binary designs, there exist tokens too weak for the small model yet unnecessary for the large model, leading to redundant recomputation. This motivates our research question: rather than merely optimizing the drafter or the verifier, *can we design a hierarchical verification framework that to handle medium-confidence tokens?*

We first verify this motivation through theoretical analysis. By aligning the distributions of the drafter and the verifier into a common space, we establish that the rejection rate—which directly determines the speedup—is correlated with the distance between the two distributions. When this distance is measured using KL divergence, its non-Euclidean property (Kullback & Leibler, 1951) guarantees the existence of an intermediate distribution that makes the two models closer through it (§3. Hence, the idea of introducing an intermediate layer is theoretically grounded.

We further practice by proposing a three-tier paradigm : Hierarchical Verification(HVSD), see Figure 1). Specifically, we introduce a slim-verifier constructed by skipping layers from the large model. In HVSD, tokens drafted by the small model are first verified in parallel by the slim-verifier; if a token is rejected, the s li m decides whether to regenerate it on its own (when sufficiently confident) or defer it to the large model (when uncertain). This design effectively reduces the invocation frequency of the large model while maintaining quality. We validate the effectiveness of this design across multiple models and demonstrate that it consistently

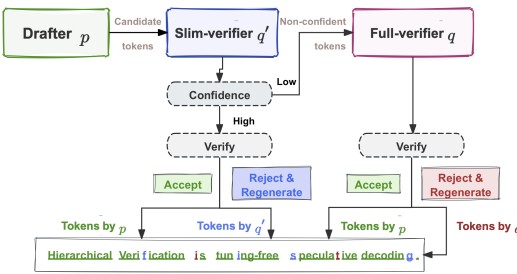

Figure 1: Illustration of our pipeline.

achieves higher speedups compared to the traditional two-tier draft–verify framework. Specifically, Our contributions are:

**(i)** We prove from an information-theoretic perspective that KL divergence provides a principled basis for introducing an intermediate distribution into the draft–verify mechanism, thereby validating the feasibility of breaking the two-tier paradigm (§4.1).

**(ii)** We design training-free Slim-verifier, a skip-layer intermediate verifier that approximates the large model through Dynamic layer-skipping adaptation (DLSA), efficiently handling medium-confidence tokens and reducing redundant full-model calls (§4.2).

**(iii)** We propose Hierarchical Verification for Speculative Decoding (HVSD) as a general three-tier framework that integrates drafter, slim-verifier, and full verifier, fundamentally overcoming the binary limitations of draft–verify methods (§4.3).

**(iv)** Through experiments of summarization, translation, reasoning, QA, and coding tasks, HVSD achieves consistently lower rejection rates (0.1–0.22) and delivers 10–20% speedup over state-of-the-art speculative decoding baselines(§5).

## 2 RELATED WORK

**Drafting Strategies.** Drafting methods fall into two categories: independent and adaptive drafters. Independent drafters employ small or non-autoregressive models (Xia et al., 2023; Leviathan et al., 2023), including SpecInfer (Miao et al., 2023), Sequoia (Chen et al., 2024), OPT-Tree (Wang et al., 2024), and DSBD (Qin et al., 2024). They are simple but often yield low acceptance due to distribution mismatch across tiers. Adaptive drafters reuse the target model's structure, either by adding feed-forward heads for parallel generation (Stern et al., 2018; Cai et al., 2024) or by layer skipping and early exiting to form lightweight submodels (Zhang et al., 2023; Yang et al., 2023). Medusa (Cai et al., 2024) improves throughput with multi-head drafting, while Eagle (Li et al., 2024) introduces residual interactions between draft and verification.

**Verification Strategies.** Verification has traditionally relied on strict, lossless checks (Stern et al., 2018; Xia et al., 2023), ensuring equivalence to $q$ but incurring rollback overhead. Lossy verification (Leviathan et al., 2023; Zhou et al., 2023) relaxes acceptance rules, tolerating bounded divergence across tiers to improve efficiency. Cascades (Chen et al., 2025b; Narasimhan et al., 2025) reframes cascading as target-distribution selection, embedding deferral policies within SD. Token tree verification (Miao et al., 2023; Cai et al., 2024; Li et al., 2024) further parallelizes the process by validating multiple candidate paths.

## 3 PRELIMINARIES

### 3.1 DRAFT–VERIFY MECHANISM VIA DISTRIBUTION DISTANCE

**Draft–Verify.** The key idea of speculative decoding is a "draft–verify" mechanism (Stern et al., 2018; Leviathan et al., 2023; Xia et al., 2023): a small model $p$ drafts candidate tokens, and a large model $q$ verifies them in parallel. Let the vocabulary be $V$ and the set of probability distributions over it

be $\Delta_V$. Given a prefix $x_{<t}$, the drafter $p$ and the verifier $q$ define conditional distributions at step $t$: $p_t(\cdot \mid x_{<t})$, $q_t(\cdot \mid x_{<t})$. Let the draft block size be $\gamma$. During the drafting phase (first tier), the small model sequentially generates $\gamma$ candidate tokens: $x_t, x_{t+1}, \ldots, x_{t+\gamma-1} \sim p_t, p_{t+1}, \ldots, p_{t+\gamma-1}$. In the verification phase (second tier), the large model simultaneously computes its distributions for the same prefixes: $\{q_t, \ldots, q_{t+\gamma-1}\}$, and checks whether the drafted tokens should be accepted or replaced. When decoding, if the first rejection occurs at position $j^*$, decoding halts at $x_{t+j^*}$ and resumes from this new prefix. If all candidates are accepted, multiple tokens are output in one step.

In the general abstraction of lossy speculative decoding (Tran-Thien, 2023), the decision consists of two parts. First, the acceptance probability for a drafted token is defined as $\Pr[\text{accept } x_{t+j}] = \min\left(1, \frac{q_{t+j}(x_{t+j})}{(1-\alpha)p_{t+j}(x_{t+j})}\right)$, where $\alpha \in [0,1)$ is a threshold parameter. If the candidate is rejected, the token is redrawn from the *residual distribution* defined as $x_{t+j} \sim \text{norm}\left(\max\left\{0, \frac{1}{\beta} q_{t+j}(\cdot) - p_{t+j}(\cdot)\right\}\right)$, where $\beta \geq 1 - \alpha$ is a scaling parameter and $\text{norm}(\cdot)$ normalizes a non-negative vector into a probability distribution. This ensures that replacements remain aligned with $q$'s distribution. In the special case $\alpha = 0, \beta = 1$, the process becomes equivalent to autoregressive sampling from $q$, *i.e.*, a lossless setting (Leviathan et al., 2023; Chen et al., 2023).

The per-step rejection rate is then defined by the probability that a drafted token from $p_t$ is not accepted by $q_t$. Formally, this can be expressed as

$$\rho_t(\alpha) = \sum_{v \in V} \max\{0, \, q_t(v) - (1-\alpha)p_t(v)\}. \tag{1}$$

**Distance Perspective.** In the lossless setting ($\alpha = 0, \beta = 1$), the rejection rate simplifies to

$$\rho_t = D_{\text{TV}}(p_t, q_t) \quad \Longrightarrow \quad \text{Rej}(p, q) \propto D(p, q). \tag{2}$$

$D_{\text{TV}}(\cdot)$ here means total variation distance for details. This formulation highlights that the rejection rate is essentially a function of the distributional distance between $p$ and $q$. In other words, the rejection rate can be naturally interpreted from a *distance perspective*: **the smaller the distance between $p$ and $q$, the lower the rejection rate.**

> **Insight 1.** *Hence, our broader objective is to explore: What kinds of distribution distances between $p$ and $q$ can effectively reduce the rejection rate? By reconstructing the distributions in an appropriate space and modeling their distance, we may uncover a principled path to reducing rejection rates.*

### 3.2 RECONSTRUCTING THE DISTANCE IN Π-SPACE

Measuring the distance directly in the original distribution space can be too coarse and may fail to reveal the underlying structure. To address this issue, we introduce a target distribution Π, within which the relationship between $p$ and $q$ can be reconstructed under a more general framework.

**Lossy Variant in Π-Space.** Formally, the lossy speculative decoding rule can be rewritten as

$$\pi_t(v) := (1 - \delta)\, p_t(v) + \delta\, q_t(v). \tag{3}$$

where $\pi_t$ denotes the target distribution at time step $t$, and $\delta \in [0, 1]$ is a weighting parameter. Here, $\Pi = \{\pi_t\}_{t=1}^{L}$ represents the collection of distributions across the entire sequence, while $\pi_t$ is the local projection at each step. In other words, $\Pi$ is the global object and $\pi_t$ its per-step realization.

*The problem then becomes:* how should we measure the divergence between $p$ and $q$ in $\pi$-space so as to accurately reflect the dynamics of the rejection rate? To this end, we consider three classical measures of distributional divergence and compare them under the same assumptions.

A natural candidate is the **Total Variation (TV) distance** (Villani, 2008):

$$D_{\text{TV}}(p, q) = \tfrac{1}{2} \sum_{v \in V} \big|p(v) - q(v)\big|. \tag{4}$$

In the lossless case ($\alpha = 0, \beta = 1$), the single-step rejection rate satisfies $\rho_t = D_{\text{TV}}(p_t, q_t)$. This reveals a strict "measure–mechanism" consistency: TV not only characterizes the geometric

discrepancy between distributions but also equals the observable rejection probability. Furthermore, due to symmetry and the triangle inequality, for any intermediate distribution $r$ it holds that $D_{\mathrm{TV}}(p, q) \leq D_{\mathrm{TV}}(p, r) + D_{\mathrm{TV}}(r, q)$. Hence, under TV, the direct path $p \to q$ is always the shortest; introducing an intermediate distribution cannot shorten the distance. In other words, if TV is adopted, reducing the rejection rate can only be achieved by directly shrinking $D_{\mathrm{TV}}(p, q)$.

We now turn to the **Kullback–Leibler (KL) divergence** (Kullback & Leibler, 1951):

$$D_{\mathrm{KL}}(p\|q) \;=\; \sum_{v \in V} p(v) \log \frac{p(v)}{q(v)}. \tag{5}$$

Unlike TV divergence, KL divergence is a Bregman divergence induced by negative entropy, and it does not satisfy symmetry or the triangle inequality. This non-Euclidean property introduces a key possibility: within certain restricted families (*e.g.*, convex feasible sets aligned with task priors, model structures, or implementation constraints), there may exist an intermediate distribution such that the "broken line path" under KL is no longer than the direct one. Formally, let $S$ be a non-empty closed convex set on the probability simplex, and define $r^* \;=\; \arg\min_{r \in S} D_{\mathrm{KL}}(p\|r)$. Then, by the generalized Pythagorean theorem in information geometry, we have

$$D_{\mathrm{KL}}(p\|q) \;\geq\; D_{\mathrm{KL}}(p\|r^*) \;+\; D_{\mathrm{KL}}(r^*\|q), \qquad \forall q \in S. \tag{6}$$

This inequality strictly confirms that such a possibility indeed exists: in the KL framework, a path via $r^*$ can yield a lower overall cost than the direct path. Detailed analysis of measures seeAppendix C.

> **Insight 2.** *The information projection property of KL provides a theoretical foundation for potentially reducing effective discrepancy—and thus rejection rate—through carefully constructed intermediate distributions.*

## 4 METHODOLOGY

### 4.1 THE DRAFT-SLIM-FULL VERIFICATION MECHANISM

**Theoretical: Multi-Tier Structure.** From the conclusions of Equations (5) and (6), we know that when KL divergence is adopted as the measure of discrepancy, there exists the *possibility* that a "piecewise path" is no longer worse than the direct path. To operationalize this possibility, we first specify the *candidate set* of intermediate points. Let $\mathcal{U} \subseteq \Delta_V$ denotes the feasible family of intermediate distributions, consistent with task priors, model structure, and deployment constraints (*e.g.*, GPU memory, latency, or throughput). Typical constructions of $\mathcal{U}$ include affine or moment constraints, exponential-family closures, or spans induced by a finite collection of deployable proxy models (such as series variants or layer-skipped submodels). For theoretical clarity, we assume $\mathcal{U}$ to be a non-empty closed convex set within the probability simplex.

Under this setting, information geometry ensures that for any $p \in \Delta_V$, the *information projection* onto $\mathcal{U}$ exists: $u^* \;=\; \arg\min_{u \in \mathcal{U}} D_{\mathrm{KL}}(p\|u)$, and satisfies the generalized Pythagorean relation

$$D_{\mathrm{KL}}(p\|q) \;\geq\; D_{\mathrm{KL}}(p\|u^*) \;+\; D_{\mathrm{KL}}(u^*\|q), \qquad \forall q \in \mathcal{U}. \tag{7}$$

Moreover, whenever $\mathcal{U}$ is not a singleton, there will generally exist infinitely many $u \in \mathcal{U}$ such that the "piecewise path is no longer worse than the direct one" (*e.g.*, when $\mathcal{U}$ contains an $I$-orthogonal submanifold through $u^*$, or local perturbation families around $q$). This indicates that, from a purely theoretical perspective, the set of beneficial candidates is typically not unique but infinite. Detailed analysis see Appendix D

**Practical: Three-Tier Structure.** Here, we introduce $u$ as intermediate distribution to form a three-tier structure $p - u - q$. In practice, $u$ acts as a *intermediate verifier*, verifying tokens drafted by $p$ before passing them to $q$ or fallback mechanisms. This design aims to filter or redirect uncertain tokens at low cost, thereby reducing the overall rejection and rollback burden while maintaining output quality.

Specifically, when the large model $q$ verifies the candidates drafted by the small model $p$, the acceptance condition in speculative decoding is:

$$q(x) \;\geq\; (1 - \alpha)\, p(x) \quad \Longleftrightarrow \quad \log q(x) - \log p(x) \;\geq\; \log(1 - \alpha), \tag{8a}$$

$$p(x) \;\leq\; \tfrac{1}{\beta}\, q(x) \quad \Longleftrightarrow \quad \log q(x) - \log p(x) \;\geq\; \log \beta. \tag{8b}$$

Both thresholds can thus be unified as inequalities of log-likelihood ratios. The extent to which these conditions are violated is quantified as the *margin violation*. To capture this violation, we introduce a convex positive-part function $\phi : \mathbb{R} \to \mathbb{R}_{\geq 0}$, we use $\phi(z) = \max\{0, z\}$ for a hard threshold, and $\phi(z) = \log(1 + e^z)$ for a smooth and differentiable surrogate, further discussion see Appendix F.

Based on this formulation, we define the KL-style single-step cost. For the direct case where $p$ drafts and $q$ verifies, the cost is

$$R_{\alpha,\beta}^{\mathrm{KL}}(q \parallel p) := \underbrace{\mathbb{E}_{x\sim p}\left[\phi\left(\log \frac{(1-\alpha)\,p(x)}{q(x)}\right)\right]}_{\text{log-margin violation of acceptance threshold}} + \underbrace{\mathbb{E}_{x\sim q}\left[\phi\left(\log \frac{\beta\,p(x)}{q(x)}\right)\right]}_{\text{log-margin contribution of residual replacement}} . \quad (9)$$

The first term penalizes cases where $q$'s support for $p$ falls short of the $(1-\alpha)$ threshold, while the second term accounts for the log-margin cost of residual replacement, *i.e.*, when $q$ must "borrow" probability mass to replace $p$'s draft. In an extended three-tier structure, if $u$ first verifies $p$, we similarly obtain $R_{\alpha,\beta}^{\mathrm{KL}}(u\|p)$; and if $q$ subsequently verifies $u$, we obtain $R_{\alpha,\beta}^{\mathrm{KL}}(q\|u)$. Thus, each segment of the three-tier pipeline can be formalized through the same log-margin perspective.

To align with sequence generation, this single-step cost must be accumulated across the decoding time steps $\mathcal{T}$. In the $\Pi$-space induced by the lossy mechanism in Equation (3), the block-level cost is

$$C_{\alpha,\beta}^{\mathrm{KL}}(q\|p \mid \pi) = \sum_{t\in\mathcal{T}} R_{\alpha,\beta}^{\mathrm{KL}}\big(q_t\|p_t\big), \quad q_t, p_t \in \Delta_V \text{ induced by } \pi. \quad (10)$$

Here, $\pi$ ensures consistency among the conditional distributions $\{p_t, q_t, u_t\}$ across different layers. However, in the definition of $C_{\alpha,\beta}^{\mathrm{KL}}(q\|p \mid \pi)$, the expectations involved cannot be computed analytically during inference, and thus must be transformed into a computable form. For example, let $\ell^p = \log p_t(\cdot)$ and $\ell^q = \log q_t(\cdot)$, and define $z_1(v) = \log(1-\alpha) + \ell^p(v) - \ell^q(v)$, $z_2(v) = \log\beta + \ell^p(v) - \ell^q(v)$. When choosing $\phi(z) = \max\{0, z\}$ (*i.e.*, ReLU), the KL-style lossy cost for a single decoding step can be written as

$$R_{\alpha,\beta}^{\mathrm{KL}}(q\|p)\big|_t = \underbrace{\sum_v p_t(v)\,\mathrm{ReLU}\big(z_1(v)\big)}_{\text{acceptance threshold term}} + \underbrace{\sum_v q_t(v)\,\mathrm{ReLU}\big(z_2(v)\big)}_{\text{residual replacement term}} . \quad (11)$$

This expression converts the original log-margin violation and residual replacement requirements into a token-wise computable sum. By further summing over all decoding steps $t \in \mathcal{T}$, the block-level cost is obtained as $C_{\alpha,\beta}^{\mathrm{KL}}(q\|p \mid \pi) = \sum_{t\in\mathcal{T}} R_{\alpha,\beta}^{\mathrm{KL}}(q\|p)\big|_t$, thereby making Equation (10) computable in practice. The detailed derivation is provided in Appendix G.

On this basis, we naturally define the discriminant between direct and three-tier paths. For the direct path $p \to q$, the cost is $C_{\alpha,\beta}^{\mathrm{KL}}(q\|p|\pi)$. For the folded path $p \to u \to q$, the total cost is the sum of its two segments, namely $C_{\alpha,\beta}^{\mathrm{KL}}(u\|p|\pi) + C_{\alpha,\beta}^{\mathrm{KL}}(q\|u|\pi)$. Their difference gives

$$\Delta_{\alpha,\beta}^{\mathrm{KL}}(u \mid \pi) = C_{\alpha,\beta}^{\mathrm{KL}}(q\|p \mid \pi) - \Big(C_{\alpha,\beta}^{\mathrm{KL}}(u\|p \mid \pi) + C_{\alpha,\beta}^{\mathrm{KL}}(q\|u \mid \pi)\Big). \quad (12)$$

The quantity $\Delta_{\alpha,\beta}^{\mathrm{KL}}(u|\pi)$ serves as the key criterion for evaluating the benefit of introducing an intermediate verifier: (1) When the value is positive, the folded path is strictly superior under the lossy rule and $\pi$-space, meaning that introducing $u$ significantly reduces the total log-margin violation. (2) When it is 0, the folded path is equivalent to the direct path; (3) When negative, the candidate $u$ should be discarded (see Figure 2). Hence, we can extend the original two-tier definition of the target distribution $\pi$ to a three-tier version that explicitly incorporates the intermediate verifier. Formally, the target distribution in the presence of $u$ is $\pi_t^{(u)}(v) := (1-\delta_2)\big((1-\delta_1)\,p_t(v) + \delta_1\,u_t(v)\big) + \delta_2\,q_t(v)$, with $\delta_1, \delta_2 \geq 0$. The coefficients are induced by the lossy thresholds $(\alpha, \beta)$ and reflect the proportions of tokens routed through each stage.

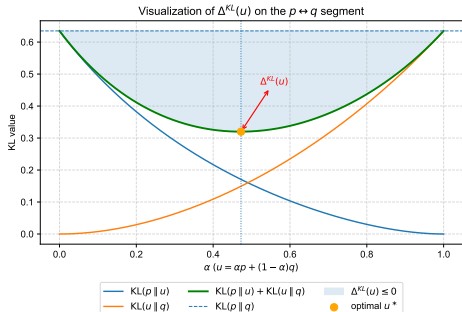

Figure 2: Visualization of $\Delta^{KL}(u)$ in Equation (12), where the shaded region shows $\Delta^{KL}(u)! \leq! 0$ that yields a shorter divergence and the orange marker the optimal $u^*$.

## 4.2 CONSTRUCTING THE TRAINING-FREE SLIM-VERIFIER

To further operationalize, we first need to clarify the possible candidates for $u$. Broadly speaking, they can be categorized into three types: (1) Scale-up version $p'$ of the drafter $p$, (2) A smaller model from the same series as the verifier $q$, and (3) A Submodel $q'$ of the verifier $q$. Here we choose skip-layer model $q'$. From a theoretical perspective, the skip-layer submodel $q'$ is constructed to have the same parameter scale as a same-family intermediate model, yet its behavior under the KL-based criterion is fundamentally different. When we substitute $u = q'$ into the cost formulation Equation (11), the block-level term $C_{\alpha,\beta}^{\mathrm{KL}}(q\|q' \mid \pi)$ remains significantly smaller than $C_{\alpha,\beta}^{\mathrm{KL}}(q\|u \mid \pi)$ for the independent intermediate model of equal size. This advantage arises because $q'$ shares embeddings and the output head with the large model $q$, thereby preserving distributional consistency that independent models cannot match. Consequently, even at the same scale, the folded path $p \to q' \to q$ is more likely to satisfy $\Delta_{\alpha,\beta}^{\mathrm{KL}}(u \mid \pi) > 0$, leading to a tangible reduction in rejection rates. Under moderate skip ratios, this enables a controllable trade-off between acceleration and accuracy (see Figure 3 and Appendix E). Therefore, we adopt $q'$ as the intermediate point $u$ in our framework.

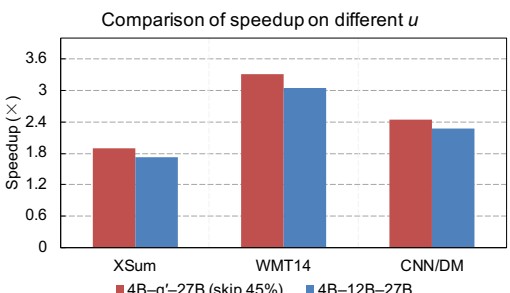

Figure 3: Skip-layer intermediate consistently outperforms the independent 12B model on Gemma3 pilot study.

**Skip-Layer Model as slim-verifier.** Let the full verifier $q$ consist of $L$ Transformer layers. We define a skipping set as $z \in \{0, 1\}^L$, where $z_\ell = 1$ indicates that the $\ell$-th layer is retained, and $z_\ell = 0$ indicates that it is skipped. Based on $z$, the slim-verifier can be formally defined as $q_z'(x) = \Pi_z(q)(x)$, where $\Pi_z(\cdot)$ denotes the projection of $q$ onto the subspace specified by $z$, resulting in a compact submodel $q_z'$.

For each candidate skipping set $z$ that defines a slim-verifier $q_z'$, we score $z$ by accumulating a KL-style per-step cost, see Equation (11), over a context window of length $\tau$:

$$\mathcal{C}(z) = \sum_{t=1}^{\tau} R_{\alpha,\beta}^{\mathrm{KL}}(q \| q_z')\Big|_t, \quad z^* = \arg\min_z \mathcal{C}(z). \tag{13}$$

To balance efficiency and accuracy, DLSA combines *random search* with *periodic Bayesian optimization*:

$$z = \begin{cases} \mathrm{BayesOpt}(l), & \text{if } o \bmod \theta = 0, \\ \mathrm{RandomSearch}(l), & \text{otherwise,} \end{cases} \tag{14}$$

where $o$ is the current optimization step, $\theta$ is the triggering period for Bayesian optimization, and $l = \binom{L}{rL}$ represents the search space of candidate skipping sets (with $r$ denoting the skip ratio). This hybrid strategy enables efficient exploration while progressively approaching the optimal skipping configuration. The structure of $q'$ is fixed once either the maximum optimization steps $S$ is reached, or the best candidate remains unchanged across multiple iterations. At this point, $q'$ serves as a stable approximation of $q$, preserving its output distribution while significantly reducing computational cost.

## 4.3 HIERARCHICAL VERIFICATION PIPELINE

At each decoding cycle beginning at step $t$, the drafter $p$ proposes a *block* of up to $\gamma$ tokens, denoted $\tilde{x}_{t:t+\gamma-1} \sim p$. This blockwise Under a fixed skipping ratio $r$, we then obtain the slim-verifier $q'$ from the full verifier $q$ via *Dynamic Layer-Skipping Adaptation* (DLSA); its distribution is denoted by $q_t'(v)$. The DLSA procedure performs dynamic optimization over a context window so that $q'$ becomes a stable approximation of $q$ that can be incorporated into the verification pipeline.

In the verification stage, two thresholds $(\delta_1, \delta_2)$ regulate the contributions of the two verification layers: the $p \to q'$ gate is set more strictly (via $\delta_1$) so that only candidates sufficiently consistent with $q'$ are allowed to pass early, while the $u \to q$ gate is set more leniently (via $\delta_2$) so that more candidates reach the final check by $q$. In practice, $\delta_1 \gg \delta_2$: a larger $\delta_1$ enforces strict early filtering that relies on $q'$, whereas a smaller $\delta_2$ prevents excessive rejection at the final stage and stabilizes overall performance.

---

**Algorithm 1** Hierarchical Verification with DLSA (Simplified; full version in Algorithm 2)

---

**Require:** Drafter $p$, full verifier $q$, skip ratio $r$, block length $\gamma$, thresholds $(\alpha_1, \alpha_2)$
**Ensure:** Final decoded sequence $x$
 1: **while** not end-of-sequence **do**
 2:     Draft a block $\tilde{x}_{t:t+\gamma-1} \sim p$
 3:     Build slim-verifier $q'$ from $q$ via DLSA under ratio $r$
 4:     Accept the longest prefix $\kappa$ under the $\alpha$-gated rules for $(p \to q')$ and $(q' \to q)$ (cf. Eq. (15))
 5:     **if** $\kappa < \gamma$ **then**
 6:         Fallback to $q$ at $t + \kappa$
 7:     **end if**
 8:     Append accepted tokens;    $t \leftarrow t + \kappa$
 9: **end while**

---

Operationally, the two-layer verification consumes the drafted block $\tilde{x}_{t:t+\gamma-1}$ and accepts a longest valid prefix of length

$$\kappa = \max_{0 \le k \le \gamma} \left\{ k \mid \tilde{x}_{t:t+k-1} \in \mathcal{A}(p \to q', \, q' \to q) \right\}, \tag{15}$$

where $\mathcal{A}(\cdot)$ denotes the acceptance set of prefixes that successfully pass the verifiers. If $\kappa < \gamma$, the pipeline falls back to $q$ at step $t + \kappa$ to continue decoding, ensuring correctness while retaining most of the speedup from the accepted prefix.

In this process, the thresholds $\alpha_1$ and $\alpha_2$ define the confidence gates at different layers and are mapped to $\delta_1$ and $\delta_2$ for mixture weights. Specifically, $\alpha_1$ determines whether the slim-verifier $q'$ accepts the drafter $p$'s token, with the criterion $q'_t(v) \ge (1 - \alpha_1) \max_u q'_t(u)$,. Meanwhile, $\alpha_2$ decides whether $q'$ is insufficiently confident and thus requires escalation to the full verifier $q$, i.e., $q$ is invoked when $q'_t(v) < (1 - \alpha_2) \max_u q'_t(u)$,. A larger $\delta_1$ (stricter $\alpha_1$) ensures that only tokens highly consistent with $q'$ are accepted early, while a smaller $\delta_2$ (looser $\alpha_2$) lowers escalation frequency and stabilizes performance. At the token level, the output distribution is

$$\pi_t^{(q')}(v) \;=\; (1 - \delta_2)\left((1 - \delta_1)\, p_t(v) \;+\; \delta_1\, q'_t(v)\right) \;+\; \delta_2\, q_t(v), \tag{16}$$

where $\delta_1 = \mathbb{P}[\, q'_t(v) \ge (1 - \alpha_1) \max_u q'_t(u)\,]$ and $\delta_2 = \mathbb{P}[\, q'_t(v) < (1 - \alpha_2) \max_u q'_t(u)\,]$, with the detailed derivation deferred to Appendix B.

## 5 EXPERIMENTS

### 5.1 EXPERIMENTAL SETTING

**Setup and Metrics.** We evaluate our framework across two major model families: **T5** (Raffel et al., 2020)and **Gemma**(Team et al., 2024), covering different model scales to examine both scalability and generality. For T5, we use *T5-small → T5-large* and *T5-large → T5-XL* as drafter–verifier pairs. For Gemma, we study *Gemma2-2B → Gemma2-9B* and *Gemma2-2B → Gemma2-27B*. This selection allows us to compare performance both within the encoder–decoder (T5) and decoder-only (Gemma) architectures. Evaluation is conducted on a diverse set of benchmarks: XSum (Narayan et al., 2018) and CNN/DailyMail summarization (Nallapati et al., 2016), WMT14 En–De translation (Bojar et al., 2014) on T5. GSM8K (8-shot reasoning) (Cobbe et al., 2021), MBPP (code generation) (Austin et al., 2021), SQuAD 2.0 (Rajpurkar et al., 2016), WebQuestions (Berant et al., 2013), NaturalQA (Kwiatkowski et al., 2019), and TriviaQA (Joshi et al., 2017) on Gemma. We report both quality metrics (ROUGE-1/2/L, BLEU for summarization and translation; EM/F1 for QA; pass@k for coding tasks; and accuracy for reasoning) and efficiency metrics (acceptance rate, rollback ratio, and relative latency speedup over greedy decoding). All experiments are performed on A100 GPUs with consistent batch size and decoding configurations to ensure comparability.

**Baselines.**    To comprehensively evaluate our method, we compare it against six representative baselines spanning both independent drafting and self-drafting approaches. Independent drafting methods include Speculative Decoding (Leviathan et al., 2023) and BiLD (Kim et al., 2023a), which

Table 1: Comparison of speculative decoding and related methods. Our approach introduces an intermediate skip-layer verifier $u$, extending the traditional two-tier SD into a three-tier structure.

| Category | Method | Drafting / Verification Source | Lossless / Lossy | Decision |
|---|---|---|---|---|
| Independent drafting | Speculative Decoding (Leviathan et al., 2023) | Small model $p$ | Lossless | $\alpha(x) = \min\left(1, \frac{q(x)}{p(x)}\right)$ |
| | BiLD (Kim et al., 2023b) | Internal little decoder of $q$ | Lossless | $\mathbf{1}[\max_x p(x) < \tau]$ |
| | Cascade SD (Chen et al., 2025b) | Small model $p_1, p_2$ | Lossy | $\mathbf{1}[\max_x p(x) < \tau]$ |
| | Faster Cascades (Narasimhan et al., 2025) | Small model $p$ | Lossy | $\pi(x) = T(p, q), \ \mathbf{1}[\pi(x) < \tau]$ |
| Self-drafting | Swift (Xia et al., 2025) | Skip-layer submodel $q'$ | Approx. Lossless | $\alpha(x) = \min\left(1, \frac{q(x)}{p'(x)}\right)$ |
| | CLaSP (Chen et al., 2025a) | Dynamic skip-layer submodel $q'$ | Approx. Lossless | $\alpha(x) = \min\left(1, \frac{q(x)}{p'(x)}\right)$ with dynamic layers |
| Ours | Skip-layer Verification | Drafter $p$ + skip-layer verifier $q'$ | Lossy | $\alpha_{q'}(x) = \min\left(1, \frac{q'(x)}{p(x)}\right)$; if reject $\to q$ |

employ either an external small model or an internal lightweight decoder. We also consider lossy baselines such as Cascade Speculative Drafting (Chen et al., 2025b) and Faster Cascades (Narasimhan et al., 2025), which improve throughput by relaxing verification rules or embedding deferral into the speculative decoding pipeline. In contrast, self-drafting methods like Swift (Xia et al., 2025) and CLaSP (Chen et al., 2025a) leverage skip-layer submodels derived from the target model, reducing distribution mismatch without extra training.

**Hyperparameters.** For the setting of $alpha$, We set $\alpha_1 = 0.5$ and $\alpha_2 = 0.3$, which consistently reduces rejection while avoiding unnecessary calls to $q$. Besides, we set $\gamma = 5$ as block size for maximum drafting length, and 45% skip-ratio to form the slim-verifier.

## 5.2 MAIN RESULTS

Table 2: Performance comparison (Xsum, CNNDM, and WMT14) under different decoding settings.

| Model | Methods | XSum (T=1) | | | CNNDM (T=1) | | | WMT14 (T=0) | | |
|---|---|---|---|---|---|---|---|---|---|---|
| | | ROUGE-2 | Rejection Rate | Speedup | ROUGE-2 | Rejection Rate | Speedup | BLEU | Rejection Rate | Speedup |
| T5 S-L | Speculative Decoding | 14.80 | 0.36 | 1.02× | 11.20 | 0.36 | 1.79× | 18.00 | 0.30 | 1.40× |
| | BiLD | 14.80 | 0.34 | 1.10× | 11.20 | 0.34 | 1.85× | 18.00 | 0.29 | 1.55× |
| | Cascade SD | 14.80 | 0.33 | 1.15× | 11.20 | 0.33 | 1.90× | 18.00 | 0.28 | 1.60× |
| | Faster Cascades | 15.05 | 0.30 | 1.30× | 12.63 | 0.34 | 1.88× | **22.65** | 0.25 | 1.85× |
| | SWiFT | 14.80 | 0.32 | 1.20× | 11.20 | 0.32 | 1.92× | 18.00 | 0.27 | 1.70× |
| | CLaSP | 14.80 | 0.31 | 1.25× | 11.20 | 0.31 | 1.95× | 18.00 | 0.26 | 1.75× |
| | Ours | **15.27** | **0.24** | **1.90×** | **12.81** | **0.26** | **2.10×** | 21.92 | **0.22** | **2.50×** |
| T5 S-XL | Speculative Decoding | 18.90 | 0.38 | 1.12× | 12.90 | 0.38 | 1.70× | 22.95 | 0.34 | 1.80× |
| | BiLD | 18.90 | 0.36 | 1.20× | 12.90 | 0.36 | 1.82× | 22.95 | 0.33 | 1.95× |
| | Cascade SD | 18.90 | 0.35 | 1.25× | 12.90 | 0.35 | 1.85× | 22.95 | 0.32 | 2.00× |
| | Faster Cascades | **18.99** | 0.30 | 1.45× | 12.95 | 0.34 | 1.95× | 23.05 | 0.25 | 2.70× |
| | SWiFT | 18.90 | 0.34 | 1.30× | 12.90 | 0.34 | 1.95× | 23.00 | 0.31 | 2.20× |
| | CLaSP | 18.90 | 0.33 | 1.35× | 12.90 | 0.33 | 2.00× | 23.00 | 0.30 | 2.25× |
| | Ours | 18.95 | **0.28** | **1.65×** | **13.05** | **0.24** | **2.40×** | **23.10** | **0.21** | **3.35×** |

**T5 Models.** In the T5 experiments, the target verifier $q$ is instantiated as T5-large or T5-XL. Independent speculative decoding uses T5-small or T5-large as drafter $p$. For Swift and CLaSP baselines, the drafter $p'$ is obtained by skipping layers only in the decoder of $q$, with full encoder preserved. Swift adopts a fixed skip ratio, whereas CLaSP applies dynamic layer selection during decoding. In our method, we retain whole as the drafter and use skip-layer decoder of large model, with encoder remains intact as in $q$ model. As shown in Table 2, under greedy decoding ($t = 0$), our method reduces rejection rates to 0.20–0.22, significantly outperforming Faster Cascades (0.31–0.32) and Swift/CLaSP (0.34–0.36), while achieving 2.5–3.3× speedup, about 30% higher than existing methods. The generation quality (ROUGE-2 / BLEU) remains consistent with the large model. Under sampling decoding ($t = 1$), although the gap narrows, our method still leads with rejection rates of 0.24–0.28 and speedups of 2.8–3.1×, whereas other methods typically exceed 0.33 rejection with speedups below 2.5×. Overall, our approach consistently achieves the lowest rejection rates and highest speedups across model scales and decoding settings, demonstrating the advantage of hierarchical verification in balancing efficiency and stability. Detailed experiments see Appendix H .

**Gemma Models.** For the Gemma experiments, we adopt Gemma2-2B $\to$ 9B and Gemma2-2B $\to$ 27B as drafter–verifier pairs. Unlike T5, Gemma follows a decoder-only architecture, which simplifies speculative decoding to a direct two-tier (or three-tier in our case) pipeline. We evaluate on GSM8K (8-shot) for reasoning, MBPP for coding, and QA benchmarks (SQuAD 2.0, WebQuestions,

NaturalQA, TriviaQA). Quality is measured by task-specific metrics (accuracy, F1, EM, pass@k), while efficiency is measured by acceptance and relative latency. As shown in Table 3, our method demonstrates advantages across multiple dimensions on the WebQuestions, NaturalQA, and TriviaQA datasets. For Gemma2-2b/9b, on WebQuestions our approach achieves the lowest rejection rate (0.10) with a speedup of $1.82\times$; on NaturalQA it again yields the lowest rejection rate (0.17) and a $2.10\times$ speedup; and on TriviaQA it obtains the lowest rejection rate (0.13) along with the highest speedup of $2.33\times$. For the larger Gemma2-2b/27b setting, the trend remains consistent: while baseline Speculative Decoding reaches a rejection rate as high as 0.45 on NaturalQA, our method reduces it to 0.30, and achieves superior latency across all datasets (up to $2.61\times$ and $2.50\times$). Detailed experiments refer to Appendix H.

Table 3: Performance comparison of results on WebQuestions, NaturalQA, and TriviaQA.

| Model | Baseline | WebQuestions | | | NaturalQA | | | TriviaQA | | |
|---|---|---|---|---|---|---|---|---|---|---|
| | | Accuracy | Rejection Rate | Speedup | Accuracy | Rejection Rate | Speedup | Accuracy | Rejection Rate | Speedup |
| Gemma2 2b,9b | Speculative Decoding | 0.27 | 0.17 | 1.50× | 0.26 | 0.27 | 1.32× | 0.50 | 0.21 | 1.55× |
| | BiLD | 0.27 | 0.15 | 1.59× | 0.26 | 0.24 | 1.67× | 0.50 | 0.19 | 1.85× |
| | Cascade SD | 0.27 | 0.16 | 1.53× | 0.26 | 0.26 | 1.43× | 0.50 | 0.20 | 1.62× |
| | Faster Cascades | 0.28 | 0.13 | 1.65× | 0.27 | 0.22 | 1.75× | 0.52 | 0.17 | 1.90× |
| | SWiFT | 0.27 | 0.15 | 1.59× | 0.26 | 0.25 | 1.61× | 0.50 | 0.18 | 1.73× |
| | CLaSP | 0.27 | 0.14 | 1.62× | 0.26 | 0.24 | 1.67× | 0.50 | 0.17 | 1.90× |
| | **Ours** | **0.28** | **0.10** | **1.82×** | **0.28** | **0.17** | **2.10×** | **0.53** | **0.13** | **2.33×** |
| Gemma2 2b,27b | Speculative Decoding | 0.32 | 0.27 | 1.55× | 0.32 | 0.45 | 1.54× | 0.54 | 0.24 | 1.65× |
| | BiLD | 0.32 | 0.26 | 1.63× | 0.32 | 0.41 | 1.92× | 0.54 | 0.22 | 2.15× |
| | Cascade SD | 0.32 | 0.24 | 1.71× | 0.32 | 0.42 | 1.73× | 0.54 | 0.23 | 1.81× |
| | Faster Cascades | **0.33** | 0.22 | 1.81× | 0.33 | 0.40 | 2.10× | **0.56** | 0.20 | 2.30× |
| | SWiFT | 0.32 | 0.21 | 1.89× | 0.32 | 0.41 | 1.93× | 0.54 | 0.23 | 1.82× |
| | CLaSP | 0.32 | 0.20 | 1.97× | 0.32 | 0.39 | 2.21× | 0.54 | 0.22 | 1.99× |
| | **Ours** | 0.32 | **0.14** | **2.32×** | **0.34** | **0.30** | **2.61×** | 0.55 | **0.15** | **2.50×** |

**Ablation.** Our ablations reveal complementary roles of the three hyper-parameters. First, the skip-ratio achieves the best trade-off around 40–50%, where speedup peaks while rejection remains controlled. Second, the draft length $\gamma$ exhibits a unimodal effect: too small underutilizes batching, while too large increases rollbacks; the optimal lies at $\gamma \approx 5$ for smaller models and 6–7 for larger ones. Finally, the thresholds $(\alpha_1, \alpha_2)$ govern efficiency–quality balance: $\alpha_1 \approx 0.5$ and a moderate $\alpha_2$ (0.3–0.4) consistently yield the highest or near-highest acceleration across datasets. Together, these results indicate that moderate settings across all three dimensions provide robust and general-purpose configurations. Ablation brief of skip-ratio is shown in Figure 4. More ablation results are given in Appendix H.

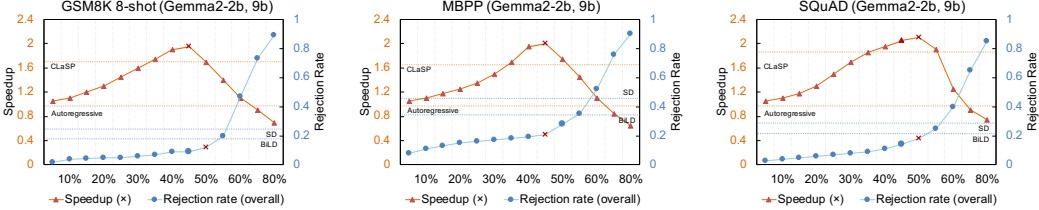

Figure 4: Ablation result of skip-ratio on Gemma2-2b,9b.

## 6 CONCLUSION

This paper proposed a three-tier speculative decoding framework with hierarchical verification, where a training-free skip-layer submodel serves as a slim-verifier to bridge the gap between draft and large models. This design lowers rejection rates and achieves more stable speedups without sacrificing quality or training. However, the current approach is limited in that the slim-verifier is directly derived from the large model without task-specific adaptation. In future work, we would explore learning-based or adaptive slim-verifiers, extend the framework to multi-modal generation, and study its robustness under diverse tasks and deployment constraints.

**Ethics Statement** This work focuses on decoding efficiency for pre-trained language models and does not involve the collection of new human subjects data. All datasets used in evaluation are publicly available under their respective licenses, and we comply with the terms of use for each dataset. The tasks include QA and reasoning benchmarks that may contain distributional biases; our methods are evaluated holistically across multiple datasets to reduce overfitting to any single source. The proposed techniques optimize inference efficiency without altering the underlying training data or optimizing model parameters, thereby not introducing additional data-related risks. We do not intentionally process or store personally identifiable information (PII). Potential misuse risks are similar to those of baseline models; we encourage responsible deployment, including appropriate content filtering and monitoring consistent with community standards.

**Reproducibility Statement** We aim to make all results reproducible. Our supplementary materials provide: the exact model variants evaluated; dataset splits and preprocessing steps; decoding settings (e.g., temperature, top-$k$/$p$, maximum lengths), verifier call patterns, and any acceptance/rejection rules; and the software environment (framework versions and key dependencies). Upon publication, we will release code and instructions to enable faithful replication and extension of our findings.

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

# APPENDIX

## TABLE OF CONTENTS

# A COMPLIANCE AND ADDITIONAL STATEMENTS

## A.1 USAGE OF LARGE LANGUAGE MODELS

Large Language Models (LLMs) were employed solely for writing assistance and linguistic refinement. Specifically, they were used to enhance clarity, readability, and overall fluency by supporting tasks such as sentence rephrasing, grammar correction, and stylistic polishing.

Importantly, LLMs were **not involved** in the conception of research ideas, methodological design, or experimental analysis. All scientific contributions—including the development of research questions, design of experiments, and interpretation of results—were independently carried out by the authors. The use of LLMs was limited strictly to improving the linguistic presentation of the manuscript.

The authors take full responsibility for the entire content of the paper, including any text that may have been refined with LLM assistance. All generated or polished text complies with academic integrity standards and does not constitute plagiarism or scientific misconduct. Accordingly, our disclosure for ICLR 2026 is: *"Yes, to aid or polish writing. Details are described in the paper."*

## A.2 ETHICS STATEMENT

This work focuses on decoding efficiency for pre-trained language models and does not involve the collection of new human subjects data. All datasets used in evaluation are publicly available under their respective licenses, and we comply with the terms of use for each dataset. The tasks include QA and reasoning benchmarks that may contain distributional biases; our methods are evaluated holistically across multiple datasets to reduce overfitting to any single source. The proposed techniques optimize inference efficiency without altering the underlying training data or optimizing model parameters, thereby not introducing additional data-related risks. We do not intentionally process or store personally identifiable information (PII). Potential misuse risks are similar to those of baseline models; we encourage responsible deployment, including appropriate content filtering and monitoring consistent with community standards.

## A.3 REPRODUCIBILITY STATEMENT

We aim to make all results reproducible. Our supplementary materials provide: the exact model variants evaluated; dataset splits and preprocessing steps; decoding settings (e.g., temperature, top-$k/p$, maximum lengths), verifier call patterns, and any acceptance/rejection rules; and the software environment (framework versions and key dependencies). Upon publication, we will release code and instructions to enable faithful replication and extension of our findings.

# B    HIERARCHICAL VERIFICATION PIPELINE: DERIVATION OF $\delta_1, \delta_2$ AND THEIR RELATION TO $\alpha_1, \alpha_2$

## B.1    FULL PSEUDOCODE FOR PIPELINE

---

**Algorithm 2** Hierarchical Verification with DLSA (aligned with Section 4.3)

---

**Require:** Drafter $p$, full verifier $q$, skip ratio $r$, block length $\gamma$, thresholds $(\alpha_1, \alpha_2)$
**Ensure:** Final decoded sequence $x$
1: Initialize running estimates for mixture weights $\delta_1, \delta_2$ (optional)
2: **while** not end-of-sequence **do**
3:      Draft a block $\tilde{x}_{t:t+\gamma-1} \sim p$
4:      Build slim-verifier $q'$ from $q$ via DLSA under ratio $r$
5:      Run $q'$ in parallel on prefixes $x_{<t}, \ldots, x_{<t+\gamma-1}$ to obtain $q'_{t:t+\gamma-1}(\cdot)$
6:      $j^* \leftarrow$ None                                   ▷ earliest rewrite/reject position
7:      **for** $j = 0$ **to** $\gamma - 1$ **do**
8:          $v \leftarrow \tilde{x}_{t+j}, \quad M' \leftarrow \max_u q'_{t+j}(u)$
9:          **if** $q'_{t+j}(v) \geq (1 - \alpha_2) M'$ **then**              ▷ $q'$ is confident (no escalation)
10:             **if** $q'_{t+j}(v) \geq (1 - \alpha_1) M'$ **then**            ▷ $q'$ accepts $p$'s token
11:                 accept $v$                                ▷ keep $\tilde{x}_{t+j}$
12:                 optionally update $\delta_1 \leftarrow$ running-avg$\big[\mathbf{1}\{q'_{t+j}(v) \geq (1-\alpha_1)M'\}\big]$
13:             **else**
14:                 rewrite $x_{t+j} \sim q'_{t+j}$                     ▷ $q'$ replaces the token
15:                 $j^* \leftarrow j$; **break**                ▷ earliest change triggers rollback
16:             **end if**
17:          **else**                                    ▷ $q'$ not confident $\Rightarrow$ escalate to $q$
18:             invoke $q$ on prefix $x_{<t+j}$; let $M \leftarrow \max_u q_{t+j}(u)$
19:             **if**  $q$ accepts $v$ under the chosen rule (lossless or $\pi$-based)  **then**
20:                 accept $v$
21:             **else**
22:                 rewrite $x_{t+j}$ using $q$ (or residual)
23:                 $j^* \leftarrow j$; **break**
24:             **end if**
25:             optionally update $\delta_2 \leftarrow$ running-avg$\big[\mathbf{1}\{q'_{t+j}(v) < (1-\alpha_2)M'\}\big]$
26:          **end if**
27:      **end for**
28:      **if** $j^* =$ None **then**
29:          $\kappa \leftarrow \gamma$                                     ▷ whole block accepted
30:      **else**
31:          $\kappa \leftarrow j^*$                              ▷ longest valid prefix length
32:          fallback to $q$ at position $t + \kappa$                        ▷ as in Eq. (15)
33:      **end if**
34:      append the accepted prefix $x_{t:t+\kappa-1}$ to $x$; $\quad t \leftarrow t + \kappa$
35: **end while**
   *Note:* The analytical mixture $\pi_t^{(q')}(v)$ in Eq. (16) uses $\delta_1 = \mathbb{P}[\, q'_t(v) \geq (1 - \alpha_1) \max_u q'_t(u)\,]$ and $\delta_2 = \mathbb{P}[\, q'_t(v) < (1 - \alpha_2) \max_u q'_t(u)\,]$ (Appendix B.2); it does not alter the control flow above.

---

## B.2    PROBABILISTIC DERIVATION OF $\delta_1, \delta_2$

At decoding step $t$, let the drafter distribution be $p_t(\cdot)$, the pre-verifier distribution be $q'_t(\cdot)$, and the full verifier distribution be $q_t(\cdot)$. We denote $M'_t = \max_{u \in V} q'_t(u)$. Given thresholds $0 < \alpha_2 < \alpha_1 < 1$, define $\tau_1 = 1 - \alpha_1, \qquad \tau_2 = 1 - \alpha_2, \qquad$ with $\tau_1 > \tau_2$. We distinguish three mutually exclusive

events (relative to a drafted token $v$):

$$A_t(v) := \{\, q'_t(v) \geq \tau_1 M'_t \,\} \qquad\qquad (q' \text{ confident and } \textbf{accepts } p\text{'s token}),$$
$$B_t(v) := \{\, \tau_2 M'_t \leq q'_t(v) < \tau_1 M'_t \,\} \qquad (q' \text{ confident but } \textbf{rewrites } \text{with } q'),$$
$$C_t(v) := \{\, q'_t(v) < \tau_2 M'_t \,\} \qquad\qquad (q' \text{ not confident} \Rightarrow \textbf{escalate } \text{to } q).$$

Clearly $A_t(v) \cup B_t(v) \cup C_t(v) = \Omega$.

**(1) Theoretical Analysis (Full Probability Decomposition).** Define the marginal probabilities under $v \sim p_t$:

$$\delta_2 := \mathbb{P}_{v \sim p_t}[\, C_t(v)\,],$$
$$\bar{\delta}_1 := \mathbb{P}_{v \sim p_t}[\, B_t(v)\,],$$
$$\delta_p := \mathbb{P}_{v \sim p_t}[\, A_t(v)\,].$$

Since $\delta_p + \bar{\delta}_1 + \delta_2 = 1$, the output distribution becomes

$$\pi_t(v) = \delta_p\, p_t(v) + \bar{\delta}_1\, q'_t(v) + \delta_2\, q_t(v). \tag{A.1}$$

Equivalently, one can express the weights via indicator expectations:

$$\delta_p = \sum_{u \in V} p_t(u)\, \mathbf{1}\{q'_t(u) \geq \tau_1 M'_t\}, \tag{A.2}$$

$$\bar{\delta}_1 = \sum_{u \in V} p_t(u)\, \mathbf{1}\{\tau_2 M'_t \leq q'_t(u) < \tau_1 M'_t\}, \tag{A.3}$$

$$\delta_2 = \sum_{u \in V} p_t(u)\, \mathbf{1}\{q'_t(u) < \tau_2 M'_t\}. \tag{A.4}$$

This yields the three-way mixture of *accept with $p$*, *rewrite with $q'$*, and *escalate to $q$*. We refer to Equation (A.1)- A.4 as the **theoretical analysis version**.

**(2) Practical Version (Confidence-Gated Mixture; aligned with main text).** In practice, we use confidence gates determined by $\alpha_1, \alpha_2$ and define the mixture weights as *unconditional* probabilities under $v \sim p_t$:

$$\delta_1 := \mathbb{P}_{v \sim p_t}\big[q'_t(v) \geq (1 - \alpha_1) \max_u q'_t(u)\big], \qquad \delta_2 := \mathbb{P}_{v \sim p_t}\big[q'_t(v) < (1 - \alpha_2) \max_u q'_t(u)\big]. \tag{A.5}$$

Introduce the Bernoulli gates

$$I_{\text{conf}}(v) = \mathbf{1}\{\, q'_t(v) \geq (1 - \alpha_1) \max_u q'_t(u)\,\}, \qquad I_{\text{esc}}(v) = \mathbf{1}\{\, q'_t(v) < (1 - \alpha_2) \max_u q'_t(u)\,\}.$$

Conditioned on these gates, the instantaneous output at step $t$ is

$$\pi_t(v \mid I_{\text{conf}}, I_{\text{esc}}) = \big(1 - I_{\text{esc}}(v)\big)\Big(\big(1 - I_{\text{conf}}(v)\big)p_t(v) + I_{\text{conf}}(v)\, q'_t(v)\Big) + I_{\text{esc}}(v)\, q_t(v).$$

Taking expectation over $v \sim p_t$ and the gates yields the confidence-gated mixture

$$\pi_t(v) = (1 - \delta_2)\Big((1 - \delta_1)\, p_t(v) + \delta_1\, q'_t(v)\Big) + \delta_2\, q_t(v), \tag{A.6}$$

which matches the main-text Equation (16). In words, a larger $\delta_1$ (induced by a stricter $\alpha_1$) increases the weight of $q'$ within the non-escalation branch, and a smaller $\delta_2$ (from a looser $\alpha_2$) reduces escalation to $q$, stabilizing performance.

*Remark (normalized conditional alternative).* If one prefers a branch-normalized view, define

$$\widetilde{\delta}_1 := \mathbb{P}_{v \sim p_t}\big[q'_t(v) \geq (1 - \alpha_1) \max_u q'_t(u) \mid q'_t(v) \geq (1 - \alpha_2) \max_u q'_t(u)\big] = \frac{\mathbb{P}[B_t(v)]}{1 - \delta_2},$$

and replace $\delta_1$ in (A.4) by $\widetilde{\delta}_1$; the functional form is unchanged, and both parameterizations are equivalent up to re-scaling within the non-escalation branch.

This conditional form corresponds to the implementation in Algorithm 2. We refer to Equation (A.5)- Equation (A.6) as the **practical version**, which matches the main-text Equation (16).

**Relation to $\alpha_1, \alpha_2$.** With $\tau_1 = 1 - \alpha_1$, $\tau_2 = 1 - \alpha_2$, the three regions correspond to:

- **Acceptance region** $A_t$: $q'_t(v) \geq (1 - \alpha_1) \max_u q'_t(u)$,
- **Rewrite region** $B_t$: $(1 - \alpha_2) \max_u q'_t(u) \leq q'_t(v) < (1 - \alpha_1) \max_u q'_t(u)$,
- **Escalation region** $C_t$: $q'_t(v) < (1 - \alpha_2) \max_u q'_t(u)$.

> **Insight (learnable thresholds).** While in the main analysis we treat $\alpha_1$ and $\alpha_2$ as **fixed hyper-parameters**, one may also consider optimizing them directly during training or tuning. In this view, $\alpha_1, \alpha_2$ can be parameterized as learnable gates, with gradients obtained from the mixture distribution $\pi_t(v)$ in Equation (A.6). Such an approach allows the system to adaptively calibrate the strictness of the confidence and escalation thresholds, potentially yielding better trade-offs between accuracy and efficiency compared to manually chosen constants.

### B.3 ANALYSIS OF THRESHOLD PARAMETERS $\alpha_1, \alpha_2$

We analyze how the two thresholds control routing and, consequently, the reliance on the large model $q$. Recall that the pre-verifier $q'$ accepts a drafted token $v$ from $p$ when

$$q'_t(v) \geq (1 - \alpha_1)\max_u q'_t(u),$$

and escalation to $q$ occurs when

$$q'_t(v) < (1 - \alpha_2)\max_u q'_t(u),$$

with the token-level mixture $\pi_t^{(q')}(v) = (1 - \delta_2)\big((1 - \delta_1)p_t(v) + \delta_1 q'_t(v)\big) + \delta_2 q_t(v)$ where $\delta_1 = \Pr\big[q'_t(v) \geq (1 - \alpha_1) \max_u q'_t(u)\big]$, $\delta_2 = \Pr\big[q'_t(v) < (1 - \alpha_2) \max_u q'_t(u)\big]$.

**Monotonicity.** Define the *reliance on $q$* at step $t$ as the escalation probability $\mathcal{R}_t := \delta_2$. Because the event set $E_{\alpha_2} = \{ q'_t(v) < (1 - \alpha_2) \max_u q'_t(u) \}$ shrinks as $\alpha_2$ increases (the threshold $(1 - \alpha_2)$ decreases), we have

$$\alpha_2 \uparrow \quad \Rightarrow \quad E_{\alpha_2} \downarrow \quad \Rightarrow \quad \mathcal{R}_t = \Pr(E_{\alpha_2}) \text{ is non-increasing.}$$

Thus, larger $\alpha_2$ *reduces* escalation (smaller $\delta_2$) and lowers dependence on $q$; smaller $\alpha_2$ does the opposite.

For the acceptance gate, the event $A_{\alpha_1} = \{ q'_t(v) \geq (1 - \alpha_1) \max_u q'_t(u) \}$ *expands* as $\alpha_1$ increases, hence

$$\alpha_1 \uparrow \quad \Rightarrow \quad A_{\alpha_1} \uparrow \quad \Rightarrow \quad \delta_1 = \Pr(A_{\alpha_1}) \text{ is non-decreasing.}$$

Therefore, larger $\alpha_1$ makes $q'$ *less* strict (more tokens accepted by $q'$ rather than rewritten or escalated), while smaller $\alpha_1$ enforces stricter early filtering.

**Implications for efficiency/quality.** Let $c_p, c_{q'}, c_q$ be the per-token inference costs of $p, q', q$ with $c_p \ll c_{q'} < c_q$. Ignoring rollbacks for clarity, the expected per-token cost satisfies

$$\mathbb{E}[C_t] \approx (1 - \delta_2)\big((1 - \delta_1)c_p + \delta_1 c_{q'}\big) + \delta_2 c_q.$$

Hence,

$$\frac{\partial \mathbb{E}[C_t]}{\partial \alpha_2} = \frac{\partial \mathbb{E}[C_t]}{\partial \delta_2} \cdot \frac{\partial \delta_2}{\partial \alpha_2} \leq \big(c_q - ((1 - \delta_1)c_p + \delta_1 c_{q'})\big) \cdot 0 \leq 0,$$

since $\partial \delta_2 / \partial \alpha_2 \leq 0$. Increasing $\alpha_2$ lowers cost (improves speed) by suppressing escalation to $q$; decreasing $\alpha_2$ trades speed for quality by invoking $q$ more often. Similarly,

$$\frac{\partial \mathbb{E}[C_t]}{\partial \alpha_1} = \frac{\partial \mathbb{E}[C_t]}{\partial \delta_1} \cdot \frac{\partial \delta_1}{\partial \alpha_1} = \big(c_{q'} - c_p\big) \cdot \frac{\partial \delta_1}{\partial \alpha_1} \geq 0,$$

so larger $\alpha_1$ (looser pre-verification) slightly *increases* the share handled by $q'$ (vs. $p$), raising cost mildly; smaller $\alpha_1$ (stricter) pushes more tokens away from $q'$ (either rewritten less or escalated), which can decrease cost but may increase rollback/escalation rates depending on $\alpha_2$.

## C   COMPARISON OF DIVERGENCE MEASURES IN $\Pi$-SPACE

In the main text we introduced the $\Pi$-space formulation and highlighted the role of divergence measures in connecting rejection rate with geometric discrepancy between $p$ and $q$. Here we provide a more complete comparison of several widely used distances.

**Total Variation (TV) distance.**
$$D_{\text{TV}}(p, q) \;=\; \tfrac{1}{2} \sum_{v \in V} |p(v) - q(v)|.$$

TV is symmetric and satisfies the triangle inequality. Moreover, in the lossless case ($\alpha = 0$), the rejection rate coincides exactly with $D_{\text{TV}}(p, q)$. This tight consistency makes TV appealing, but also restrictive: since the direct path $p \to q$ is always shortest, introducing intermediate distributions cannot reduce the effective distance. Thus, TV provides a faithful but rigid geometry where rejection rate is tightly locked to the global discrepancy.

**Kullback–Leibler (KL) divergence.**
$$D_{\text{KL}}(p\|q) \;=\; \sum_{v \in V} p(v) \log \frac{p(v)}{q(v)}.$$

KL is asymmetric and does not satisfy the triangle inequality. Instead, it admits a non-Euclidean structure induced by negative entropy. This allows the possibility of information projection: for a convex feasible set $S$, one can find
$$r^* = \arg\min_{r \in S} D_{\text{KL}}(p\|r),$$
and the generalized Pythagorean theorem guarantees
$$D_{\text{KL}}(p\|q) \;\geq\; D_{\text{KL}}(p\|r^*) + D_{\text{KL}}(r^*\|q), \qquad \forall q \in S.$$
Hence, unlike TV, an intermediate distribution $r^*$ can strictly reduce the effective discrepancy. This property provides the theoretical foundation for reducing rejection rate by designing suitable verification pathways.

**Jensen–Shannon (JS) divergence.**
$$D_{\text{JS}}(p\|q) \;=\; \tfrac{1}{2} D_{\text{KL}}\big(p\big\|\tfrac{p+q}{2}\big) + \tfrac{1}{2} D_{\text{KL}}\big(q\big\|\tfrac{p+q}{2}\big).$$
JS is symmetric and bounded, and its square root defines a metric. However, due to its averaging nature, JS tends to blur directional information and does not directly correspond to rejection probabilities. Thus, while JS is useful for visualization and bounded analysis, it lacks operational meaning for speculative decoding.

**Wasserstein distance.**
$$W(p, q) \;=\; \inf_{\gamma \in \Gamma(p,q)} \mathbb{E}_{(u,v) \sim \gamma}[\, d(u, v)\,],$$
where $\Gamma(p, q)$ is the set of couplings of $p$ and $q$, and $d(\cdot, \cdot)$ is a ground metric. Wasserstein distance captures geometry over the support of distributions and is popular in generative modeling. However, computing it is significantly more expensive, and the link to rejection rate in speculative decoding is indirect.

**Why choose KL over TV?**   To summarize:

- TV provides exact rejection consistency but forbids improvement via intermediate distributions; KL relaxes geometric constraints, allowing the introduction of an intermediate $r^*$ to lower effective discrepancy.

- This flexibility is critical in hierarchical or lossy pipelines, where rejection dynamics depend not only on global discrepancy but also on the design of intermediate verification layers.

> **Insight.** *TV yields a faithful but rigid measure of rejection (cf. Equation (4)), whereas KL offers a more flexible geometry that enables effective discrepancy reduction via intermediate distributions (cf. Equation (5)). This property motivates our choice of KL as the primary divergence measure in $\Pi$-space.*

# D  FAMILIES OF INTERMEDIATE DISTRIBUTIONS AND MINIMAL PIECEWISE KL PATHS

## D.1  AN INFINITE FAMILY OF BENEFICIAL INTERMEDIATES

Recall the feasible family of intermediate distributions $\mathcal{U} \subseteq \Delta_V$, which is consistent with task priors, model structure, and deployment constraints. Typical constructions include affine/moment constraints, exponential-family closures, or spans induced by deployable proxy models (e.g., series variants or layer-skipped submodels). For theoretical clarity, we assume $\mathcal{U}$ to be a non-empty closed convex set.

**Lemma A.1 (Infinitely many beneficial intermediates).**  Fix $p, q \in \Delta_V$ and a non-singleton closed convex set $\mathcal{U}$ that contains $q$. Define the beneficial set

$$\mathcal{U}_{\text{ben}} = \Big\{ u \in \mathcal{U} \,\Big|\, D_{\text{KL}}(p\|q) \geq D_{\text{KL}}(p\|u) + D_{\text{KL}}(u\|q) \Big\}.$$

Then $\mathcal{U}_{\text{ben}}$ is typically infinite. In particular, let $u^* = \arg\min_{u \in \mathcal{U}} D_{\text{KL}}(p\|u)$ be the $I$-projection of $p$ onto $\mathcal{U}$. Whenever $\mathcal{U}$ contains an $I$-orthogonal submanifold through $u^*$ (or nontrivial local perturbation families around $q$), every point $u$ on such structures belongs to $\mathcal{U}_{\text{ben}}$.

*Sketch.* By the generalized Pythagorean theorem on $I$-projections, for all $q \in \mathcal{U}$,

$$D_{\text{KL}}(p\|q) \geq D_{\text{KL}}(p\|u^*) + D_{\text{KL}}(u^*\|q).$$

$u^*$ need not be unique globally if $\mathcal{U}$ admits $I$-orthogonal directions through $u^*$, and local deformations preserving $q \in \mathcal{U}$ generate continua of $u$ with the same (or smaller) two-segment cost. Hence $\mathcal{U}_{\text{ben}}$ is generically infinite. □

The lemma formalizes that the theoretical improvement via a piecewise path is *not* tied to a unique $u$: there usually exist infinitely many $u \in \mathcal{U}$ that are beneficial, which motivates designing *families* of intermediates in hierarchical pipelines rather than relying on a single pivot.

## D.2  SHORTEST PIECEWISE PATH VIA A CHAIN OF INTERMEDIATES

The KL divergence is not a metric; thus "shortest path" needs an operational definition. We define the *piecewise KL action* of a chain as the additive cost of its segments: for a sequence $p = u_0, u_1, \ldots, u_n, u_{n+1} = q$ with $u_i \in \mathcal{U}$,

$$\mathcal{L}_{\text{KL}}(u_{0:n+1}) := \sum_{i=0}^{n} D_{\text{KL}}(u_i \,\|\, u_{i+1}).$$

Our goal is to construct chains that minimize $\mathcal{L}_{\text{KL}}$ under modeling constraints.

**Theorem A.2 (Successive $I$-projections yield a minimal chain).**  Let $\mathcal{U}_0 \supseteq \mathcal{U}_1 \supseteq \cdots \supseteq \mathcal{U}_n \supseteq \{q\}$ be nested non-empty closed convex sets in $\Delta_V$. Define the successive $I$-projections

$$u_1 = \arg\min_{u \in \mathcal{U}_1} D_{\text{KL}}(u_0\|u), \quad u_2 = \arg\min_{u \in \mathcal{U}_2} D_{\text{KL}}(u_1\|u), \quad \ldots, \quad u_n = \arg\min_{u \in \mathcal{U}_n} D_{\text{KL}}(u_{n-1}\|u),$$

with $u_0 = p$ and $u_{n+1} = q$. Then the generalized Pythagorean equalities telescope to give

$$D_{\text{KL}}(p\|q) = \sum_{i=0}^{n} D_{\text{KL}}(u_i\|u_{i+1}) = \mathcal{L}_{\text{KL}}(u_{0:n+1}). \tag{A.7}$$

Moreover, among all chains that respect the same nest $\{\mathcal{U}_i\}$, the successive $I$-projection chain minimizes the piecewise action $\mathcal{L}_{\text{KL}}$.

*Sketch.* Each step satisfies $D_{\text{KL}}(u_{i-1}\|q) = D_{\text{KL}}(u_{i-1}\|u_i) + D_{\text{KL}}(u_i\|q)$ for $q \in \mathcal{U}_i$ by $I$-projection orthogonality; summing over $i$ gives (A.7). Any alternative $u_i' \in \mathcal{U}_i$ increases the corresponding segment cost by the optimality of the $I$-projection, hence increases the total action. □

**Corollary A.3 (Geodesic limit).**  Suppose $\{\mathcal{U}_i\}$ are chosen so that the successive $I$-projection chain lies on a dual-flat geodesic (e- or m-geodesic) between $p$ and $q$ in appropriate coordinates; let the mesh be refined so that $\max_i D_{\text{KL}}(u_i\|u_{i+1}) \to 0$ as $n \to \infty$. Then the polygonal chain converges to the corresponding straight line in the dual coordinates, while the piecewise action remains $D_{\text{KL}}(p\|q)$ by Equation (A.7).

**Discussion.** Theorem A.2 provides an operational notion of "shortest" under the additive KL action: a chain built from successive $I$-projections along nested convex families achieves the minimum cost, and the limit of a finely discretized chain aligns with a straight (geodesic) trajectory in the information-geometric sense. For hierarchical verification, the intermediates $u_1, \ldots, u_n$ correspond to tiers (e.g., increasingly restrictive submodels or constraints), and (A.7) explains why a well-aligned multi-tier design can realize the theoretically minimal KL action from $p$ to $q$.

### D.3 Practical Choices of $u$ in Engineering Implementation

Although $u$ can, in theory, be selected from an infinite family of intermediate distributions, in real-world system design the choice is constrained by computational cost, inference latency, hardware resources, and deployment complexity. Based on prior studies and empirical experience, the practical options for $u$ can be broadly grouped into three categories:

**(1) Scale-up of the small model** $p$**.** This approach constructs $u$ by slightly enlarging the small model $p$, for example, via LoRA, adapters, prefix-tuning, or adding a few attention or feedforward layers. The main advantages are: (i) relatively low additional inference cost due to limited model expansion, and (ii) parameter sharing with $p$, which enables rapid integration. However, the expressive power of such $u$ remains limited, and it often fails to capture the richer distributional features of $q$. As a result, its ability to reduce rejection rate is weak, especially for complex samples or long-context tasks.

**(2) Using an intermediate-size model in the same family.** Many model series (e.g., 2B–9B–27B) contain natural intermediate checkpoints. Choosing such a mid-size model as $u$ is straightforward: (i) it is generally more aligned with $q$ than $p$, providing more accurate verification across a wider range of inputs; (ii) it shares the same architecture, making it easily pluggable into the hierarchical verification pipeline. The drawbacks, however, are significant: it requires loading and maintaining a separate medium-scale model, which increases memory and bandwidth demands, complicates scheduling, and reduces system throughput. Hence, despite potential gains in accuracy, the engineering burden makes this choice less practical.

**(3) Skip-layer variant of the large model** $q$**.** A more pragmatic solution is to construct $u$ directly from $q$ by skipping certain layers or extracting lightweight sub-networks. This design offers a balance between theoretical soundness and engineering feasibility: (i) *Consistency*: since $u$ and $q$ share the same parameter space, distributional alignment is naturally preserved, avoiding model inconsistency issues; (ii) *Low overhead*: unlike introducing a standalone intermediate model, a skip-layer variant requires no additional memory footprint and only modifies inference dynamics; (iii) *Flexibility*: skipping ratios can be dynamically adjusted to trade off speed and accuracy depending on task requirements; (iv) *Stability*: empirical evidence shows that skip-layer $u$ significantly reduces rejection rate and decoding cost without sacrificing generation quality.

**Hence, we choose solution 3 as the final intermediate layer.**

# E    WHY THE DRAFTER MUST BE AN INDEPENDENT SMALL MODEL $p$ INSTEAD OF A LAYER-SKIPPED $q''$

A natural question in the hierarchical verification framework is: since the intermediate verifier $u$ can be constructed by layer-skipping from the large model $q$, why not also derive the drafter from $q$, namely a smaller $q''$, instead of using an independently trained small model $p$? From the perspective of distributional divergence, this design is problematic.

**Distribution mismatch.**    The drafter's role is to generate a large number of candidate tokens at very low cost, while maintaining a reasonable distributional consistency with $q$. This ensures both a low rejection rate and high throughput. If we attempt to derive $q''$ from $q$ via layer-skipping, forcing it to approximate the scale of $p$, we often encounter *distribution collapse*:

$$D_{\mathrm{KL}}(q \,\|\, q'') \;\gg\; D_{\mathrm{KL}}(q \,\|\, p), \tag{A.8}$$

meaning that the KL distance between $q$ and its truncated version $q''$ is significantly larger than that between $q$ and an independently trained small model $p$.

**Underlying reason.**    An independent small model $p$ is usually pretrained or distilled from larger models, which allows it to produce stable and smooth probability distributions over the vocabulary. In contrast, $q''$ is simply a *damaged copy* of $q$. Removing critical layers severely harms its representational capacity, leading to poor calibration of probabilities and distorted token likelihoods. Formally, if we write the layer-skipping projection as

$$q'' = \Pi_z(q), \qquad z \in \{0, 1\}^L,$$

where $z_\ell = 0$ indicates the $\ell$-th layer of $q$ is removed, then as $\|z\|_0$ decreases, the divergence gap

$$\Delta D \;=\; D_{\mathrm{KL}}(q\|q'') - D_{\mathrm{KL}}(q\|p) \tag{A.9}$$

tends to grow rapidly. This reflects the fact that $q''$ suffers from systematic bias rather than well-structured compression.

**Effect on rejection rate.**    In speculative decoding, the rejection rate at step $t$ is closely linked to distributional distance. In particular, under lossless conditions we have

$$\rho_t \;=\; D_{\mathrm{TV}}(p_t, q_t),$$

and under KL-style analysis, the rejection rate is upper bounded by

$$\rho_t \;\leq\; \sqrt{\tfrac{1}{2} D_{\mathrm{KL}}(p_t\|q_t)}.$$

Thus, if $p$ is replaced with $q''$, the bound is dominated by $D_{\mathrm{KL}}(q''\|q)$, which is substantially larger than $D_{\mathrm{KL}}(p\|q)$. This directly implies that the rejection rate with $q''$ would be unacceptably high.

**Practical implication.**    If $q''$ is used as the drafter, the large model $q$ would reject its outputs much more frequently, resulting in a drastically higher rejection rate that negates the acceleration benefit of speculative decoding. By contrast, a purpose-trained small model $p$ aligns better with $q$ in distribution space, leading to smaller KL divergence, stable rejection rates, and lower overall cost.

**Summary.**    Within hierarchical verification, a layer-skipped model $q''$ is a reasonable choice for the intermediate verifier $u$, but not for the drafter. The drafter must be an independent small model $p$; only this design ensures both efficiency and smoothness of distributions, avoiding KL blow-up due to distribution collapse and maintaining the acceleration gains of speculative decoding.

## F ALTERNATIVE MARGIN PENALTIES $\phi$ AND DETAILED DERIVATIONS

**Setup and unification.** Recall the acceptance inequalities (main text Eqs. (8a)–(8b)):

$$q(x) \geq (1-\alpha)\,p(x) \iff \log q(x) - \log p(x) \geq \log(1-\alpha),$$

$$p(x) \leq \tfrac{1}{\beta}\,q(x) \iff \log q(x) - \log p(x) \geq \log \beta.$$

Let $m_1 := \log(1-\alpha)$ and $m_2 := \log \beta$, and define at decoding step $t$ the logits $\ell_t^p(v) = \log p_t(v)$, $\ell_t^q(v) = \log q_t(v)$, and margins

$$z_1(v) = m_1 + \ell_t^p(v) - \ell_t^q(v), \qquad z_2(v) = m_2 + \ell_t^p(v) - \ell_t^q(v).$$

Violations correspond to $z_1(v) > 0$ (falling short of the acceptance gate) and $z_2(v) > 0$ (residual replacement pressure).

**General $\phi$-penalized single-step cost.** Let $\phi : \mathbb{R} \to \mathbb{R}_{\geq 0}$ be any convex, nondecreasing penalty ("positive-part" surrogate). A general $\phi$-style cost for one step is

$$R_{\alpha,\beta}^{\phi}(q\|p)\big|_t := \underbrace{\mathbb{E}_{v\sim p_t}\big[\phi\big(z_1(v)\big)\big]}_{\text{acceptance threshold}} + \underbrace{\mathbb{E}_{v\sim q_t}\big[\phi\big(z_2(v)\big)\big]}_{\text{residual replacement}}.$$

Expanding expectations yields a token-wise sum (computable when $p_t, q_t$ are available):

$$R_{\alpha,\beta}^{\phi}(q\|p)\big|_t = \sum_v p_t(v)\,\phi\big(z_1(v)\big) + \sum_v q_t(v)\,\phi\big(z_2(v)\big).$$

Accumulating over time $t \in \mathcal{T}$ gives the block-level cost

$$C_{\alpha,\beta}^{\phi}(q\|p\,|\,\pi) = \sum_{t\in\mathcal{T}} R_{\alpha,\beta}^{\phi}(q\|p)\big|_t, \qquad p_t, q_t \in \Delta_V \text{ induced by } \pi.$$

**Canonical choices for $\phi$ (family and properties).** We list common surrogates, all convex and nondecreasing in $z$; gradients are useful for tuning:

1. **Hinge / ReLU:** $\phi(z) = \max\{0, z\}$. Subgradient: $\phi'(z) \in [0,1]$, equals 1 for $z > 0$, 0 for $z < 0$.

2. **Squared hinge:** $\phi(z) = (z_+)^2$, $z_+ := \max\{0, z\}$. Gradient: $\phi'(z) = 2z_+\,\mathbf{1}\{z > 0\}$.

3. **Huber-hinge (parameter $\kappa > 0$):**

$$\phi_\kappa(z) = \begin{cases} 0, & z \leq 0, \\ \dfrac{z^2}{2\kappa}, & 0 < z \leq \kappa, \\ z - \dfrac{\kappa}{2}, & z > \kappa. \end{cases}$$

   Gradient: $0$, $z/\kappa$, $1$ in the three regions.

4. **Softplus / smooth hinge (temperature $\tau > 0$):** $\phi_\tau(z) = \tau \log(1 + e^{z/\tau})$. Gradient: $\phi'_\tau(z) = \sigma(z/\tau) \in (0,1)$ with $\sigma$ the logistic.
   *Bounds:* $z_+ \leq \phi_\tau(z) \leq z_+ + \tau \log 2$ for all $z$.

5. **Power hinge (aggressive):** $\phi_p(z) = (z_+)^p$, $p \geq 1$. Gradient: $\phi'_p(z) = p\,(z_+)^{p-1}\,\mathbf{1}\{z > 0\}$.

6. **Exponential positive-part (heavy-tail):** $\phi_\tau^{\exp}(z) = \max\{0, e^{z/\tau} - 1\}$, $\tau > 0$. Gradient for $z > 0$: $\phi'_\tau(z) = \frac{1}{\tau} e^{z/\tau}$; 0 for $z \leq 0$.

Choosing larger curvature (e.g., squared hinge, exponential) penalizes large violations more aggressively; smooth surrogates (softplus, Huber-hinge) provide stable gradients while remaining consistent with the hard margin in the $\tau \downarrow 0$ or $\kappa \downarrow 0$ limit.

**Ordering and calibration.** If $\phi_1(z) \leq \phi_2(z)$ for all $z$, then $R_{\alpha,\beta}^{\phi_1}(q\|p) \leq R_{\alpha,\beta}^{\phi_2}(q\|p)$ (same for $C^\phi$). In particular, with softplus temperature $\tau$,

$$R_{\alpha,\beta}^{\mathrm{ReLU}}(q\|p) \leq R_{\alpha,\beta}^{\mathrm{softplus}_\tau}(q\|p) \leq R_{\alpha,\beta}^{\mathrm{ReLU}}(q\|p) + \tau \log 2 \cdot \Big( \underbrace{\sum_v p_t(v)}_{=1} + \underbrace{\sum_v q_t(v)}_{=1} \Big) = R_{\alpha,\beta}^{\mathrm{ReLU}}(q\|p) + 2\tau \log 2.$$

**Gradients w.r.t. logits (useful for tuning).** Let $\ell_t^q(w) = \log q_t(w)$:

$$\frac{\partial}{\partial \ell_t^q(w)} \sum_v p_t(v)\, \phi\big(z_1(v)\big) = -p_t(w)\, \phi'\big(z_1(w)\big),$$

$$\frac{\partial}{\partial \ell_t^q(w)} \sum_v q_t(v)\, \phi\big(z_2(v)\big) = q_t(w)\, \phi\big(z_2(w)\big) \; - \; q_t(w)\, \mathbb{E}_{u\sim q_t}\big[\phi(z_2(u))\big] \; - \; q_t(w)\, \phi'\big(z_2(w)\big).$$

(Here we used $\partial q_t(v)/\partial \ell_t^q(w) = q_t(v)\,(\mathbf{1}\{v = w\} - q_t(w))$ and $\partial z_2(v)/\partial \ell_t^q(w) = -\mathbf{1}\{v = w\}$.) Analogous expressions hold for $\ell_t^p(w)$ if $p$ is trainable.

**Single-pass Monte Carlo via a mixture sampler.** If enumerating the vocabulary is infeasible, one may draw tokens from a mixture $\pi_t^\mu = \mu\, p_t + (1 - \mu)\, q_t$ and use importance weights:

$$\mathbb{E}_{v\sim p_t}\big[\phi(z_1(v))\big] = \mathbb{E}_{v\sim\pi_t^\mu}\left[ \frac{p_t(v)}{\pi_t^\mu(v)}\, \phi(z_1(v)) \right],$$

$$\mathbb{E}_{v\sim q_t}\big[\phi(z_2(v))\big] = \mathbb{E}_{v\sim\pi_t^\mu}\left[ \frac{q_t(v)}{\pi_t^\mu(v)}\, \phi(z_2(v)) \right].$$

A simple choice is $\mu = \frac{1}{2}$ or an adaptive $\mu$ based on acceptance rates.

**Three-tier extension.** For the folded path $p \to u \to q$, we reuse the same $\phi$-style cost on each segment: $C_{\alpha,\beta}^\phi(u\|p \,|\, \pi)$ and $C_{\alpha,\beta}^\phi(q\|u \,|\, \pi)$. The discriminant remains

$$\Delta_{\alpha,\beta}^\phi(u \,|\, \pi) = C_{\alpha,\beta}^\phi(q\|p \,|\, \pi) \; - \; \Big( C_{\alpha,\beta}^\phi(u\|p \,|\, \pi) + C_{\alpha,\beta}^\phi(q\|u \,|\, \pi) \Big),$$

which reduces to Equation (12) when $\phi = \mathrm{ReLU}$ or $\phi = \log(1 + e^z)$.

## G  FROM LLR GATES TO TOKEN-WISE $\phi$-COSTS

**Step 0: Unifying the two acceptance gates as log-margins.**  Recall the speculative-decoding acceptance conditions (Eqs. (8a)–(8b)):

$$q(x) \geq (1 - \alpha)\,p(x) \iff \log q(x) - \log p(x) \geq \log(1 - \alpha),$$

$$p(x) \leq \tfrac{1}{\beta}\,q(x) \iff \log q(x) - \log p(x) \geq \log \beta.$$

Let $m_1 := \log(1 - \alpha)$, $m_2 := \log \beta$. At decoding step $t$, define logits $\ell_t^p(v) = \log p_t(v)$, $\ell_t^q(v) = \log q_t(v)$ and the two *log-margins*

$$z_1(v) = m_1 + \ell_t^p(v) - \ell_t^q(v), \qquad z_2(v) = m_2 + \ell_t^p(v) - \ell_t^q(v).$$

Violations of the gates correspond to $z_1(v) > 0$ (falling short of the $(1-\alpha)$ acceptance) and $z_2(v) > 0$ (pressure to replace under the $1/\beta$ bound).

**Step 1: From hard constraints to a convex positive-part penalty.**  Let $\phi : \mathbb{R} \to \mathbb{R}_{\geq 0}$ be a convex, nondecreasing "positive-part" surrogate (for a hard threshold one may take $\phi(z) = \max\{0, z\}$; for a smooth version $\phi(z) = \log(1 + e^z)$, etc.). We measure the *severity* of violations by $\phi(z_1)$ and $\phi(z_2)$. Because the acceptance test in practice is *applied to candidates drafted from $p$*, its shortfall should be averaged under $p$; conversely, the *residual replacement* borrows mass from $q$, so its contribution is averaged under $q$. This yields the *single-step $\phi$-style cost*

$$R_{\alpha,\beta}^\phi(q\|p) := \mathbb{E}_{x \sim p}\left[ \phi\left( \log \frac{(1 - \alpha)\,p(x)}{q(x)} \right) \right] + \mathbb{E}_{x \sim q}\left[ \phi\left( \log \frac{\beta\,p(x)}{q(x)} \right) \right].$$

Taking $\phi = \mathrm{ReLU}$ or $\phi(z) = \log(1 + e^z)$ recovers Equation (9) in the main text.

**Step 2: Making the expectation explicit at step $t$.**  At a fixed step $t$ on a discrete vocabulary, expectations become token-wise sums:

$$\mathbb{E}_{x \sim p_t}\left[ \phi\left( \log \frac{(1 - \alpha)\,p_t(x)}{q_t(x)} \right) \right] = \sum_v p_t(v)\,\phi\big(m_1 + \ell_t^p(v) - \ell_t^q(v)\big) = \sum_v p_t(v)\,\phi\big(z_1(v)\big),$$

$$\mathbb{E}_{x \sim q_t}\left[ \phi\left( \log \frac{\beta\,p_t(x)}{q_t(x)} \right) \right] = \sum_v q_t(v)\,\phi\big(m_2 + \ell_t^p(v) - \ell_t^q(v)\big) = \sum_v q_t(v)\,\phi\big(z_2(v)\big).$$

Therefore the single-step cost is

$$R_{\alpha,\beta}^\phi(q\|p)\big|_t = \sum_v p_t(v)\,\phi\big(z_1(v)\big) + \sum_v q_t(v)\,\phi\big(z_2(v)\big). \tag{A.10}$$

**Step 3: Specializing to $\phi(z) = \max\{0, z\}$ (ReLU) gives Equation (11).**  Choosing $\phi(z) = \mathrm{ReLU}(z) = \max\{0, z\}$ in equation A.10 yields

$$R_{\alpha,\beta}^{\mathrm{KL}}(q\|p)\big|_t = \underbrace{\sum_v p_t(v)\,\mathrm{ReLU}\big(z_1(v)\big)}_{\text{acceptance threshold term}} + \underbrace{\sum_v q_t(v)\,\mathrm{ReLU}\big(z_2(v)\big)}_{\text{residual replacement term}},$$

which is exactly Equation (11) in the main text, with $z_1, z_2$ defined in equation 8.

**Step 4: Accumulating over time gives Equation (10).**  Summing the single-step costs over decoding steps $\mathcal{T}$ in the $\Pi$-space induced by the lossy rule, we obtain the block-level cost

$$C_{\alpha,\beta}^\phi(q\|p\,|\,\pi) = \sum_{t \in \mathcal{T}} R_{\alpha,\beta}^\phi(q\|p)\big|_t, \qquad p_t, q_t \in \Delta_V \text{ induced by } \pi.$$

Taking $\phi = \mathrm{ReLU}$ (or $\phi(z) = \log(1 + e^z)$) recovers Equation (10). $\qquad\square$

---

**Remarks.** *(i) The first term in equation 8 is averaged under $p$ because the acceptance gate is evaluated on drafts from $p$; the second is averaged under $q$ as it quantifies the log-margin cost when $q$ replaces $p$'s proposals.  (ii) Alternative convex surrogates (squared hinge, Huber-hinge, softplus) can replace ReLU without changing the derivation; only $\phi$ changes in equation A.10.*

## H  ADDITIONAL TABLES AND FIGURES

### H.1  ALL RESULTS ON T5 AND GEMMA

Table 4: Performance comparison (Xsum, CNNDM, and WMT14) under different decoding settings.

| Model | Methods | XSum | | | CNNDM | | | WMT14 | | |
|---|---|---|---|---|---|---|---|---|---|---|
| | | ROUGE-2 | Rejection Rate | Speedup | ROUGE-2 | Rejection Rate | Speedup | BLEU | Rejection Rate | Speedup |
| | | | | | **Greedy Decoding: Temperature=0** | | | | | |
| T5 S-L | Speculative Decoding | 16.36 | 0.33 | 1.20× | 11.00 | 0.34 | 2.05× | 18.00 | 0.39 | 1.40× |
| | BiLD | 16.36 | 0.32 | 1.25× | 11.00 | 0.33 | 2.10× | 18.00 | 0.38 | 1.55× |
| | Cascade SD | 16.36 | 0.31 | 1.30× | 11.00 | 0.32 | 2.15× | 18.00 | 0.38 | 1.55× |
| | Faster Cascades | 19.9 | 0.30 | 1.42× | 15.7 | 0.33 | 2.12× | **27.5** | 0.35 | 1.85× |
| | SWiFT | 16.36 | 0.30 | 1.35× | 11.00 | 0.31 | 2.18× | 18.00 | 0.37 | 1.70× |
| | CLaSP | 16.36 | 0.29 | 1.38× | 11.00 | 0.30 | 2.20× | 18.00 | 0.36 | 1.75× |
| | Ours | **21.3** | **0.26** | **2.15×** | **12.70** | **0.28** | **2.35×** | _21.92_ | **0.32** | **2.50×** |
| T5 S-XL | Speculative Decoding | 18.70 | 0.36 | 1.28× | 12.80 | 0.35 | 1.95× | 22.95 | 0.41 | 1.80× |
| | BiLD | 18.70 | 0.34 | 1.35× | 12.80 | 0.34 | 2.00× | 22.95 | 0.39 | 1.92× |
| | Cascade SD | 18.70 | 0.33 | 1.40× | 12.80 | 0.33 | 2.05× | 22.95 | 0.38 | 2.00× |
| | Faster Cascades | **18.90** | 0.31 | 1.60× | _12.95_ | 0.32 | 2.10× | _23.05_ | 0.31 | 2.70× |
| | SWiFT | 18.70 | 0.32 | 1.45× | 12.80 | 0.32 | 2.12× | 22.95 | 0.34 | 2.20× |
| | CLaSP | 18.70 | 0.31 | 1.48× | 12.80 | 0.31 | 2.15× | 22.95 | 0.33 | 2.25× |
| | Ours | **18.90** | **0.26** | **1.95×** | **12.99** | **0.27** | **2.75×** | **23.10** | **0.27** | **3.35×** |
| | | | | | **Non-Greedy Sampling: Temperature=1** | | | | | |
| T5 S-L | Speculative Decoding | 14.80 | 0.36 | 1.02× | 11.20 | 0.36 | 1.79× | 18.10 | 0.42 | 1.30× |
| | BiLD | 14.80 | 0.34 | 1.10× | 11.20 | 0.34 | 1.85× | 18.10 | 0.41 | 1.40× |
| | Cascade SD | 14.80 | 0.33 | 1.15× | 11.20 | 0.33 | 1.90× | 18.10 | 0.40 | 1.45× |
| | Faster Cascades | _15.05_ | 0.30 | 1.30× | _12.63_ | 0.34 | 1.88× | **22.50** | 0.36 | 1.78× |
| | SWiFT | 14.80 | 0.32 | 1.20× | 11.20 | 0.32 | 1.92× | 18.10 | 0.38 | 1.55× |
| | CLaSP | 14.80 | 0.31 | 1.25× | 11.20 | 0.31 | 1.95× | 18.10 | 0.37 | 1.60× |
| | Ours | **15.27** | **0.24** | **1.90×** | **12.81** | **0.26** | **2.10×** | _21.33_ | **0.34** | **2.30×** |
| T5 S-XL | Speculative Decoding | 18.90 | 0.38 | 1.12× | 12.90 | 0.38 | 1.70× | 23.00 | 0.44 | 1.70× |
| | BiLD | 18.90 | 0.36 | 1.20× | 12.90 | 0.36 | 1.82× | 23.00 | 0.42 | 1.80× |
| | Cascade SD | 18.90 | 0.35 | 1.25× | 12.90 | 0.35 | 1.85× | 23.00 | 0.41 | 1.85× |
| | Faster Cascades | **18.99** | 0.30 | 1.45× | _12.95_ | 0.34 | 1.95× | _23.05_ | 0.39 | 2.55× |
| | SWiFT | 18.90 | 0.34 | 1.30× | 12.90 | 0.34 | 1.95× | 23.00 | 0.40 | 2.00× |
| | CLaSP | 18.90 | 0.33 | 1.35× | 12.90 | 0.33 | 2.00× | 23.00 | 0.39 | 2.05× |
| | Ours | _18.95_ | **0.28** | **1.65×** | **13.05** | **0.24** | **2.40×** | **23.10** | **0.37** | **3.10×** |

**Hyperparameters.**  All the results confirm that with $\gamma = 5$, $\alpha_1 = 0.5$, and $\alpha_2 = 0.3$, the proposed approach outperforms all baselines in terms of the trade-off between quality, rejection rate, and speedup, thereby validating the effectiveness of the hierarchical verification design. This configuration strikes a balance between the acceptance tolerance of the intermediate verifier and the replacement flexibility of the final verifier, enabling stable comparisons across different decoding strategies.

**T5 Results Analysis.**  Table 4 reports the performance of T5 under different decoding strategies on XSum, CNNDM, and WMT14. For the smaller variant **T5-S-L**, our method achieves the best balance between quality, rejection rate, and speedup. On XSum, the ROUGE-2 score improves to 21.3 with a rejection rate reduced to 0.26, yielding a 2.15× acceleration. Similarly, on CNNDM we reach 12.7 ROUGE-2 with a rejection rate of 0.28, corresponding to a 2.35× speedup. On WMT14, our method obtains 21.92 BLEU and 2.50× speedup, surpassing all baselines. These results demonstrate that even for smaller-scale T5, hierarchical verification provides consistent gains without sacrificing quality.

For the larger **T5-S-XL**, the advantage becomes more evident. On XSum, our method reaches 18.9 ROUGE-2 with a rejection rate of 0.26, improving speedup to 1.95×, higher than Cascade SD (1.40×) and Faster Cascades (1.60×). On CNNDM, our approach delivers 12.99 ROUGE-2 and 2.75× speedup, while on WMT14 we obtain 23.10 BLEU and 3.35× speedup, marking the highest performance among all compared methods. The larger model shows both lower rejection rates and broader tolerance to longer drafts, thus benefiting more significantly from our hierarchical design.

Overall, these findings confirm that the proposed method scales effectively with model size. Compared to traditional speculative decoding and its variants, our approach reduces rejection by up to $0.1$ absolute points and increases speedup by $0.5$–$1.0\times$, while preserving or even improving generation quality.

Table 5: Performance comparison on Gemma: GSM8K, MBPP, and SQuAD 2.0.

| Model | Baseline | GSM8K | | | MBPP | | | SQuAD 2.0 | | |
|---|---|---|---|---|---|---|---|---|---|---|
| | | Quality | Rejection Rate | Latency | Quality | Rejection Rate | Latency | Quality | Rejection Rate | Latency |
| Gemma2 2b,9b | Speculative Decoding | 0.70 | 0.16 | 1.42× | 0.52 | 0.19 | 1.65× | 0.60 | 0.40 | 1.60× |
| | BiLD | 0.70 | 0.14 | 1.63× | 0.52 | 0.17 | 1.80× | 0.60 | 0.38 | 1.85× |
| | Cascade SD | 0.70 | 0.15 | 1.57× | 0.52 | 0.17 | 1.80× | 0.60 | 0.36 | 1.91× |
| | Faster Cascades | 0.72 | 0.12 | 1.82× | 0.53 | 0.16 | 1.83× | 0.60 | 0.35 | 2.05× |
| | SWiFT | 0.70 | 0.13 | 1.70× | 0.52 | 0.18 | 1.74× | 0.60 | 0.36 | 1.91× |
| | CLaSP | 0.70 | 0.12 | 1.84× | 0.52 | 0.18 | 1.75× | 0.60 | 0.34 | 2.12× |
| | Ours | **0.73** | **0.09** | **1.93×** | **0.54** | **0.12** | **1.95×** | 0.59 | **0.29** | **2.43×** |
| Gemma2 2b,27b | Speculative Decoding | 0.75 | 0.16 | 1.76× | 0.64 | 0.14 | 1.75× | 0.64 | 0.28 | 1.70× |
| | BiLD | 0.75 | 0.13 | 1.95× | 0.64 | 0.12 | 2.00× | 0.64 | 0.26 | 1.95× |
| | Cascade SD | 0.75 | 0.14 | 1.83× | 0.64 | 0.13 | 1.84× | 0.64 | 0.26 | 1.95× |
| | Faster Cascades | 0.76 | 0.10 | 2.01× | 0.65 | 0.12 | 2.03× | 0.65 | 0.25 | 2.30× |
| | SWiFT | 0.75 | 0.15 | 1.80× | 0.64 | 0.13 | 1.83× | 0.64 | 0.25 | 2.29× |
| | CLaSP | 0.75 | 0.15 | 1.81× | 0.64 | 0.12 | 2.03× | 0.64 | 0.24 | 2.47× |
| | Ours | **0.78** | **0.07** | **2.10×** | 0.63 | **0.10** | **2.20×** | **0.65** | **0.20** | **2.85×** |

| Model | Baseline | WebQuestions | | | NaturalQA | | | TriviaQA | | |
|---|---|---|---|---|---|---|---|---|---|---|
| | | Quality | Rejection Rate | Latency | Quality | Rejection Rate | Latency | Quality | Rejection Rate | Latency |
| Gemma2 2b,9b | Speculative Decoding | 0.27 | 0.17 | 1.50× | 0.26 | 0.27 | 1.32× | 0.50 | 0.21 | 1.55× |
| | BiLD | 0.27 | 0.15 | 1.59× | 0.26 | 0.24 | 1.67× | 0.50 | 0.19 | 1.85× |
| | Cascade SD | 0.27 | 0.16 | 1.53× | 0.26 | 0.26 | 1.43× | 0.50 | 0.20 | 1.62× |
| | Faster Cascades | 0.28 | 0.13 | 1.65× | 0.27 | 0.22 | 1.75× | 0.52 | 0.17 | 1.90× |
| | SWiFT | 0.27 | 0.15 | 1.59× | 0.26 | 0.25 | 1.61× | 0.50 | 0.18 | 1.73× |
| | CLaSP | 0.27 | 0.14 | 1.62× | 0.26 | 0.24 | 1.67× | 0.50 | 0.17 | 1.90× |
| | Ours | **0.28** | **0.10** | **1.82×** | **0.28** | **0.17** | **2.10×** | **0.53** | **0.13** | **2.33×** |
| Gemma2 2b,27b | Speculative Decoding | 0.32 | 0.27 | 1.55× | 0.32 | 0.45 | 1.54× | 0.54 | 0.24 | 1.65× |
| | BiLD | 0.32 | 0.26 | 1.63× | 0.32 | 0.41 | 1.92× | 0.54 | 0.22 | 2.15× |
| | Cascade SD | 0.32 | 0.24 | 1.71× | 0.32 | 0.42 | 1.73× | 0.54 | 0.23 | 1.81× |
| | Faster Cascades | **0.33** | 0.22 | 1.81× | 0.33 | 0.40 | 2.10× | **0.56** | 0.20 | 2.30× |
| | SWiFT | 0.32 | 0.21 | 1.89× | 0.32 | 0.41 | 1.93× | 0.54 | 0.23 | 1.82× |
| | CLaSP | 0.32 | 0.20 | 1.97× | 0.32 | 0.39 | 2.21× | 0.54 | 0.22 | 1.99× |
| | Ours | 0.32 | **0.14** | **2.32×** | **0.34** | **0.30** | **2.61×** | 0.55 | **0.15** | **2.50×** |

**Gemma Results Analysis.** From the results in Table 5, we set temperature $T = 1$ and observe that across both **Gemma2-2b/9b** and **Gemma2-2b/27b**, our method consistently outperforms baselines on GSM8K, MBPP, SQuAD 2.0, WebQuestions, NaturalQA, and TriviaQA. The improvements can be summarized as follows.

Overall, the differences in quality scores among methods are relatively small, indicating that most techniques primarily target efficiency. Nevertheless, our method achieves stable and sometimes notable gains. For example, on GSM8K, the quality rises from 0.70 with the baseline Speculative Decoding to 0.73 for Gemma2-2b/9b, and further to 0.78 for Gemma2-2b/27b. Similar improvements are observed on WebQuestions and NaturalQA, showing that reducing rejection does not compromise output quality, but can even enhance it.

Rejection rate is a key indicator of the strictness and stability of verification. Compared to baseline Speculative Decoding, our method substantially lowers rejection rates across tasks and scales. For instance, on NaturalQA with Gemma2-2b/27b, rejection rate drops from 0.45 to 0.30, while on TriviaQA it decreases from 0.24 to 0.15. This indicates more effective filtering of drafted candidates, leading to fewer rollbacks.

Our method consistently achieves lower latency and higher speedup, with advantages becoming more pronounced on larger models. On TriviaQA with Gemma2-2b/27b, the speedup reaches $2.50\times$, clearly surpassing Faster Cascades ($2.30\times$) and CLaSP ($1.99\times$). On NaturalQA, the maximum speedup is $2.61\times$, highlighting the method's effectiveness in mitigating inference delays and improving throughput in challenging tasks. Compared to other multi-stage speculative decoding approaches (BiLD, Cascade SD, Faster Cascades, SWiFT, and CLaSP), our method achieves a well-balanced

outcome across all three metrics: slight gains in quality, substantial reduction in rejection rate, and the highest speedup. This "triple-win" effect makes the approach highly practical, especially for latency-sensitive applications.

## H.2 ADDITIONAL ABLATION STUDIES

**Effect of $\alpha_1$ and $\alpha_2$.** Figure 5 presents the ablation results of $\alpha_1$ (strictness of rejecting the drafter $p$) and $\alpha_2$ (strictness of deferring to the full verifier $q$) on Gemma2-2b/9b. The vertical axis denotes $\alpha_1$, the horizontal axis denotes $\alpha_2$, and the color intensity reflects the relative reliance on the large model $q$. Several observations can be drawn.

First, $\alpha_2$ exerts the dominant influence. With small values of $\alpha_2$ (e.g., 0.1–0.3), the system rarely escalates to the large model, keeping reliance at a low level (around 0.1–0.2). This setting improves efficiency but risks admitting tokens that deviate from the large model's distribution, potentially degrading quality. As $\alpha_2$ increases, reliance grows significantly (above 0.6), indicating that stricter deferral triggers more frequent calls to $q$, thereby stabilizing performance at the cost of reduced speedup.

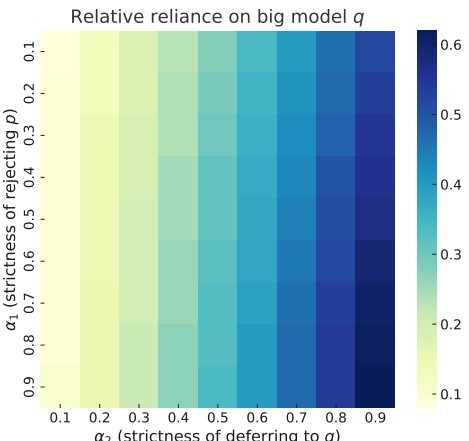
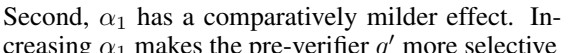

Figure 5: Ablation result of $\alpha_1$, $\alpha_2$ on Gemma2-2b,9b.

Second, $\alpha_1$ has a comparatively milder effect. Increasing $\alpha_1$ makes the pre-verifier $q'$ more selective over $p$'s tokens, modestly raising the likelihood of invoking the large model. However, the vertical variation is less pronounced than the horizontal changes driven by $\alpha_2$. This suggests that $\alpha_1$ primarily tunes the early-stage filtering, while the overall reliance is chiefly governed by $\alpha_2$.

Finally, the joint effect of $\alpha_1$ and $\alpha_2$ reveals a meaningful trade-off space. Configurations with lower $\alpha_1$ and moderate $\alpha_2$ reduce dependence on $q$ while maintaining a reasonable balance between efficiency and quality. In contrast, setting both parameters high results in near-lossless quality but almost eliminates the computational benefits.

**Ablation on $\alpha_1$ and $\alpha_2$.** The ablation study in Figure 6 investigates the impact of varying the verification thresholds $(\alpha_1, \alpha_2)$ on the overall decoding speedup. Across all six benchmarks, we observe a consistent unimodal trend with respect to $\alpha_1$: setting $\alpha_1$ too small admits overly lenient acceptance, which increases rollback overhead, while overly strict $\alpha_1$ values cause frequent intervention by the large model. The optimal balance is typically reached around $\alpha_1 \approx 0.5$, where speculative efficiency is maximized without sacrificing verification stability.

The effect of $\alpha_2$ is similarly pronounced. Relaxed values ($\alpha_2 = 0.30$ or $0.40$) yield the best speedups across most tasks, as they reduce unnecessary escalations to the large verifier. In contrast, overly strict gating ($\alpha_2 = 0.50$) consistently degrades speedup, while overly lenient gating ($\alpha_2 = 0.20$) increases rollback, both leading to suboptimal performance. Among the datasets, SQuAD2 exhibits flatter curves, suggesting higher robustness to $\alpha_2$, whereas NaturalQA and TriviaQA are more sensitive and clearly peak around $(\alpha_1 = 0.5, \alpha_2 = 0.3)$.

Overall, the ablation results highlight that the pair $(\alpha_1 = 0.5, \alpha_2 = 0.3)$ provides the most stable and effective trade-off, consistently achieving the highest or near-highest speedup across all evaluated datasets.

The ablation results of $(\alpha_1, \alpha_2)$ for Gemma2 2b→27b are presented in Figure 7. Similar to the 2b→9b setting, we observe a unimodal pattern along $\alpha_1$: both overly small and overly large values reduce efficiency, while $\alpha_1 \approx 0.5$ consistently delivers the best performance across tasks.

For $\alpha_2$, the trade-off is more pronounced. On GSM8K and SQuAD2, relaxed settings ($\alpha_2 = 0.30$ or $0.40$) achieve the highest speedups, whereas $\alpha_2 = 0.50$ significantly suppresses acceleration due to excessive verifier intervention. MBPP exhibits a slightly right-shifted optimum, with $\alpha_1 = 0.6$ paired with $\alpha_2 = 0.30/0.40$ yielding the strongest gains. NaturalQA and TriviaQA again highlight

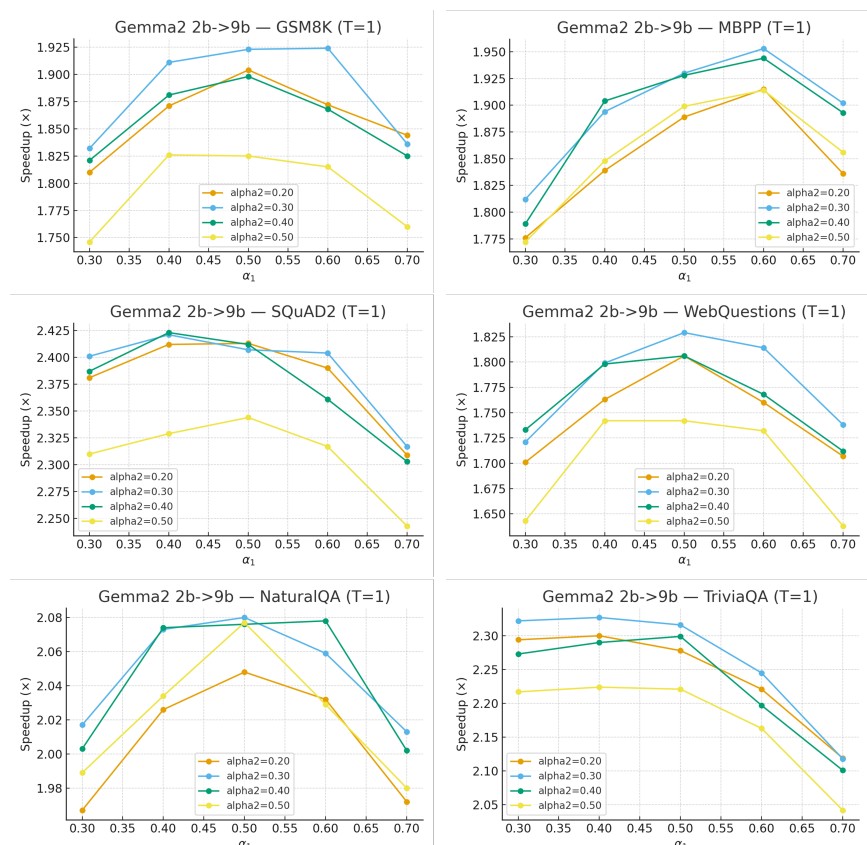

Figure 6: Ablation result of $\alpha_1$ and $\alpha_2$ on Gemma2-2b,9b.

the sensitivity to $\alpha_2$, where strict gating degrades speedup and the best region is clearly centered around $(\alpha_1 = 0.5, \alpha_2 = 0.3)$.

Overall, the 2b→27b results reinforce the robustness of $(\alpha_1 = 0.5, \alpha_2 = 0.3)$ as a general-purpose configuration, while also indicating slight task-specific variations such as MBPP's preference for a larger $\alpha_1$.

**Ablation on Draft Length $\gamma$.** To examine the effect of the draft length on acceleration, we vary the number of draft tokens $\gamma$ under fixed hyper-parameters ($\alpha_1 = 0.5$, $\alpha_2 = 0.3$, skip-ratio = 45%). As shown in Figure 8, all datasets exhibit a consistent unimodal trend: increasing $\gamma$ initially improves speedup by enlarging the expected number of accepted tokens per draft, but overly large $\gamma$ values raise the rejection probability and verification overhead, which reduces the net gain.

For **Gemma2-2b/9b**, the optimal draft length is around $\gamma = 5$. The peak speedup reaches about $2.1\times$ on GSM8K, $2.2\times$ on MBPP, and $2.6\times$ on SQuAD2, after which the curves quickly decline. This indicates that small models are more sensitive to block rejections, as rollback costs rapidly outweigh drafting benefits when $\gamma$ becomes too large.

In contrast, **Gemma2-2b/27b** shows a broader optimum around $\gamma = 6$–$7$, where the speedup rises to approximately $2.4\times$ (GSM8K), $2.6\times$ (MBPP), and nearly $3.0\times$ (SQuAD2). Even with $\gamma = 8$–$10$, the performance remains close to the maximum, reflecting that larger models generate drafts more consistent with the verifier, thus tolerating longer blocks without severe rollback penalties.

Moreover, dataset characteristics also influence the optimal $\gamma$. GSM8K, requiring long-chain reasoning, consistently yields the lowest speedup; MBPP achieves intermediate gains; while SQuAD2.0, characterized by shorter extractive outputs, attains the highest acceleration. These results suggest that $\gamma$ should be set to a moderate value ($\approx 5$ for smaller models, 6–7 for larger models) to balance draft efficiency and verification stability.

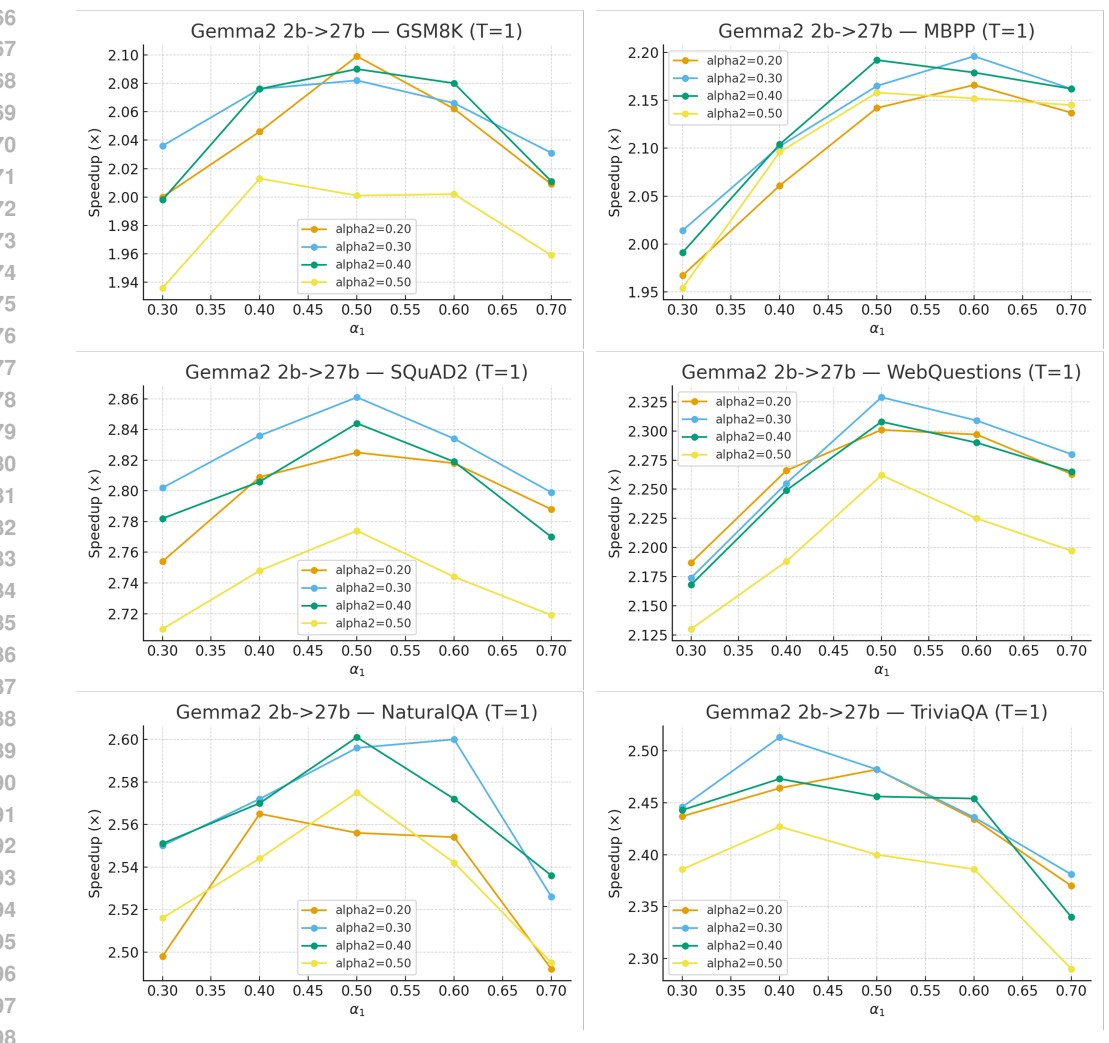

Figure 7: Ablation result of $\alpha_1$ and $\alpha_2$ on Gemma2-2b,27b.

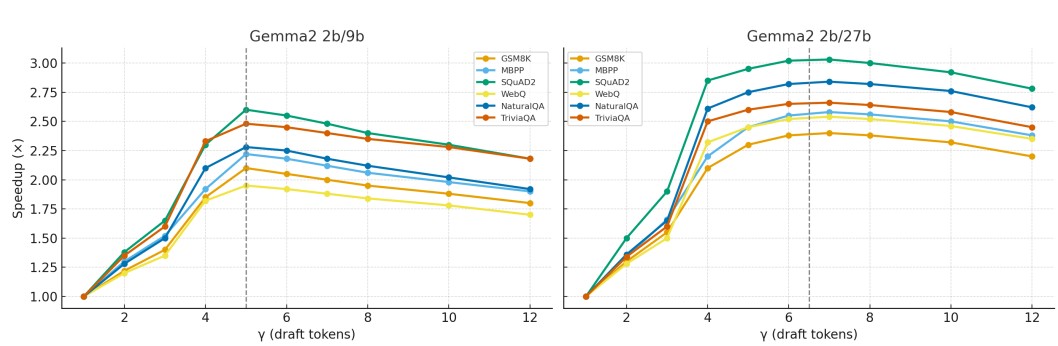

Figure 8: Ablation result of different $\gamma$ setting.

