# OpenReview forum: "Hierarchical Speculative Decoding through Training-Free Slim-Verifier"
_ICLR.cc/2026/Conference — ICLR 2026 Conference Withdrawn Submission_

### Official Review · Reviewer_XEeD · 2025-10-27

**Soundness:** 2
**Presentation:** 3
**Contribution:** 1
**Rating:** 2
**Confidence:** 4

**Summary:**

The paper proposes a three‑tier speculative decoding framework that inserts an intermediate **slim‑verifier** (denoted \(q'\)) between a small drafter \(p\) and the full target model \(q\). The slim‑verifier is constructed by skipping layers of \(q\) and is adapted at runtime via **Dynamic Layer‑Skipping Adaptation (DLSA)**. The decoding pipeline is hierarchical: tokens drafted by \(p\) are first checked by \(q'\). Accepted tokens are emitted immediately. Rejected tokens trigger one of two actions: either a local regeneration at \(q'\) when its confidence is sufficient, or a deferral to the full model \(q\) for a lossless verification step. The authors also introduce a token‑wise **lossy** acceptance gate at the intermediate tier and provide a KL‑style decomposition motivating the insertion of \(q'\). Experiments on Gemma‑2 and T5 across summarization, translation, QA, reasoning, and coding report lower rejection and \( $2\times \sim3.3\times$ \) speedups over non‑draft baselines, with gains over single‑tier SD baselines on many tasks.

**Strengths:**

1. The paper studies the observation that many rejected draft tokens have the potential to be accepted, which could reduce the expensive target model verfication.
2. The authors provide sufficent theoretical analysis to suppport their claims.
3. The presentation and the figures are clear and easy to understand.

**Weaknesses:**

1. The idea of both "multi-tier speculative decoding" and "layer-skip speculative decoding" is well explored by the related works. The motivation to accept some potentially correct draft tokens and reducing the verification cost is straightforward, but the proposed method seems more like integration of cascade and layer-skip speculative decoding methods.
2. **Theoretical gap:** KL‑projection guarantees rely on convex U, but the implemented U (skip masks) is discrete/non‑convex. The theory thus motivates but does not **justify** the claimed optimality for DLSA’s search space.
3. HVSD achieves better speedup ratio at the cost of losing the theoretical lossless property of speculative decoding, which is especially important in real-world applications.
4. Missing Related Works. Some related works [1] already explored the idea of accepting some potentially correct draft tokens. The lack of these baselines weakens the novelty and evidence.
5. The experiments and evaluation are limited to T5-series models and Gemma-series models. The authors should provide more experiments on some state-of-the-art open source LLMs, e.g. Llama 3 and Gwen 3, and extend the comparison to some latest speculative decoding methods, e.g. training-based Eagle-3 [2] and training-free PEARL [3].

[1] Bachmann, Gregor, Sotiris Anagnostidis, Albert Pumarola, Markos Georgopoulos, Artsiom Sanakoyeu, Yuming Du, Edgar Schönfeld, Ali Thabet, and Jonas Kohler. "Judge decoding: Faster speculative sampling requires going beyond model alignment." *arXiv preprint arXiv:2501.19309* (2025).

[2] Li, Yuhui, Fangyun Wei, Chao Zhang, and Hongyang Zhang. "Eagle-3: Scaling up inference acceleration of large language models via training-time test." *arXiv preprint arXiv:2503.01840* (2025).

[3] Liu, Tianyu, Yun Li, Qitan Lv, Kai Liu, Jianchen Zhu, Winston Hu, and Xiao Sun. "Pearl: Parallel speculative decoding with adaptive draft length." *arXiv preprint arXiv:2408.11850* (2024).

**Questions:**

1. Could you please provide more experiments on some extremely difficult tasks? Will HVSD significantly decrease the model performance?
2. Could you please provide measurement of the *DLSA search* time or any extra kernel launches for q′ construction? How many skip‑mask candidates are evaluated per context and per sequence? What is the wall‑clock % spent in DLSA when reporting end‑to‑end latency?
3.  KV/cache reuse. Do you share KV caches or hidden states between q′ and q? If not, what’s the measured overhead, and how would cache reuse change the results?

---

### Official Review · Reviewer_6Nir · 2025-10-31

**Soundness:** 2
**Presentation:** 2
**Contribution:** 2
**Rating:** 2
**Confidence:** 3

**Summary:**

The paper introduces HVSD, which refines the verification stage of speculative decoding into a three-tier paradigm. Specifically, the authors incorporated a slim-verifier that can directly execute accept, reject, or resample operations on tokens where it demonstrates high confidence. This design effectively minimizes the frequency of calling the full verifier.

**Strengths:**

1. The core idea of this paper holds up, and the experimental results substantiate its effectiveness.

**Weaknesses:**

1. The writing clarity of this paper is insufficient. Although I generally understood the paper's main idea, I failed to completely understand the main points that Section 3 and Section 4 attempt to explain. The author seems to be detailing the construction process of the slim-verifier $q'$. I suggest the author provide the high-level idea of this part to help understanding.
2. From a methodology perspective, the proposed HVSD is a lossy method, which approximates the target model's output by means of layer skip. Theoretically speaking, its performance (e.g., accuracy) should reasonably be lower than lossless methods (e.g., standard SD). However, the author's experimental results show that the performance of HVSD consistently outperforms standard SD. Please, the author provide a detailed explanation for this abnormal phenomenon.
3. In HVSD, for tokens where the slim-verifier confidence is lower, the full verifier is needed for processing. I want to know, if the sequence of tokens generated by the draft model contains tokens with varying confidence levels (i.e., partly high confidence, partly low confidence), how will the system handle this? Specifically, if verification is performed using a draft tree (e.g., drafting 64 tokens at once) or batching method, since the probability of all tokens being low confidence is extremely low, does this mean the full verifier will almost need to be called in every verification step? If so, this will severely impact the overall acceleration effect.
4. The proposed HVSD seems difficult to directly apply to the latest speculative decoding methods (e.g., EAGLE). In EAGLE, every draft needs to reuse the target model's information from the last verification. If the full verifier is skipped in a certain verification, then the reusable information for the next draft will be missing. Will this affect the generation quality of the next draft tokens, and further affect the overall speedup ratio?

**Questions:**

Please see the weakness.

---

### Official Review · Reviewer_HLGn · 2025-11-01

**Soundness:** 3
**Presentation:** 2
**Contribution:** 2
**Rating:** 4
**Confidence:** 4

**Summary:**

This paper introduces Hierarchical Verification for Speculative Decoding (HVSD) — a three-tier, training-free speculative decoding framework that integrates a lightweight Slim-Verifier between the drafter and the full verifier. The work aims to overcome inefficiencies in conventional two-tier draft–verify frameworks by reducing unnecessary large-model calls for tokens that can be verified by a smaller intermediate model.

**Strengths:**

- The hierarchical three-tier framework generalizes speculative decoding beyond binary verification, backed by an information-geometric justification using KL projection.
- The KL-based derivations are mathematically sound, but somewhat they took too much content in my opinion
- Practical and easy to integrate into existing inference systems.

**Weaknesses:**

- line 060: "s li m" -> "slim"
- line 315: "This blockwise Under a fixed skipping ratio r" -> what does it mean?
- The authors chose T5 as the evaluation model, this model seems a bit too old for benchmarking. Perhaps it will be better to use mainstream models like llama and qwen.
- the benchmarks tasks are allocated to these two models, instead, these benchmarks should be tested against all models.
- other methods such as EAGLE 1/2/3 should be compared?

**Questions:**

- how to tune the gate thresholds $\delta_{1}$ and $\delta_{2}$ ?
- why are the benchmark scores higher than the baselines even if the method in the paper is lossy?

---

### Note · Authors · 2025-11-13

**Comment:**

Thank you very much for the time and constructive feedback provided by the reviewers and hard work of the chairs. After careful consideration of the comments, we recognize that the current version of our manuscript is not yet sufficiently mature.

To ensure a higher-quality and more rigorous contribution, we would like to withdraw the submission at this stage.

We sincerely appreciate the reviewers’ efforts and the valuable suggestions, which will greatly help us improve our work.

**Withdrawal Confirmation:**

I have read and agree with the venue's withdrawal policy on behalf of myself and my co-authors.